# Role of p110a subunit of PI3-kinase in skeletal muscle mitochondrial homeostasis and metabolism

Mengyao Ella Li[1], Hans P.M.M. Lauritzen[1], Brian T. O'Neill[1,2], Chih-Hao Wang[1], Weikang Cai[1], Bruna B. Brandao[1], Masaji Sakaguchi[1], Rongya Tao[3], Michael F. Hirshman[1], Samir Softic[1,4] & C. Ronald Kahn[1]

Skeletal muscle insulin resistance, decreased phosphatidylinositol 3-kinase (PI3K) activation and altered mitochondrial function are hallmarks of type 2 diabetes. To determine the relationship between these abnormalities, we created mice with muscle-specific knockout of the p110α or p110β catalytic subunits of PI3K. We find that mice with muscle-specific knockout of p110α, but not p110β, display impaired insulin signaling and reduced muscle size due to enhanced proteasomal and autophagic activity. Despite insulin resistance and muscle atrophy, M-p110αKO mice show decreased serum myostatin, increased mitochondrial mass, increased mitochondrial fusion, and increased *PGC1α* expression, especially *PCG1α2* and *PCG1α3*. This leads to enhanced mitochondrial oxidative capacity, increased muscle NADH content, and higher muscle free radical release measured in vivo using pMitoTimer reporter. Thus, p110α is the dominant catalytic isoform of PI3K in muscle in control of insulin sensitivity and muscle mass, and has a unique role in mitochondrial homeostasis in skeletal muscle.

[1] Section of Integrative Physiology and Metabolism, Joslin Diabetes Center, Harvard Medical School, Boston, MA 02215, USA. [2] Division of Endocrinology and Metabolism, Fraternal Order of Eagles Diabetes Research Center, University of Iowa Carver College of Medicine, Iowa City, IA 52242, USA. [3] Division of Endocrinology, Children's Hospital Boston, Harvard Medical School, Boston, MA 02115, USA. [4] Division of Gastroenterology, Hepatology and Nutrition, Children's Hospital Boston, Harvard Medical School, Boston, MA 02115, USA. Correspondence and requests for materials should be addressed to C.R.K. (email: C.Ronald.Kahn@joslin.harvard.edu)

Phosphatidylinositol 3-kinase (PI3K) is critical in insulin action and plays key regulatory roles in cell metabolism[1]. Class IA PI3Ks are heterodimers consisting of a regulatory subunit (usually p85α/β), and a catalytic subunit (usually p110α, p110β, and p110δ) that catalyzes the production of phosphatidylinositol 3,4,5-trisphosphate (PIP3) in response to ligand stimulation[1]. Studies with tissue-specific knockout and chemical inhibitors have shown different roles of the major catalytic subunits, p110α and p110β, in insulin action in the liver and fat[2–4]. Thus, while impaired glucose tolerance is observed in mice with liver-specific knockout of either p110α or p110β, only liver-specific deletion of p110α results in a blunting of insulin-stimulated Akt phosphorylation[2,3]. Similarly, loss of p110α, but not p110β, in adipose tissue results in glucose intolerance, despite no reduction in fat mass[4]. These data indicate that while both p110α and p110β catalytic subunits are present in most insulin-sensitive tissues, in the liver and fat, p110α appears to be more important in metabolic response through the PI3K-AKT axis.

Skeletal muscle insulin resistance is a hallmark of type 2 diabetes mellitus and the obese-prediabetic state[5]. This is usually associated with mitochondrial dysfunction manifest by decreased mitochondrial content and mitochondrial oxidative capacity[6], although some studies have found no change or even increased mitochondrial metabolism in obese and insulin-resistant humans and rodents[7,8], indicating a complex relationship between mitochondrial function and insulin action. In addition, diabetes and insulin resistance have been associated with altered muscle growth and fiber type[9], and muscle wasting has been shown to correlate with defects in insulin and IGF-1 signaling in skeletal muscle[9,10]. Defects in insulin stimulation of PI3K in muscle have also been detected in type 2 diabetes, obesity, and even pre-diabetes[11,12]. The roles the p110α and p110β catalytic subunits in regulation of muscle metabolism, however, are not known.

The aim of this study was to define the relative roles of the p110α and p110β isoforms in the regulation of muscle growth and mitochondrial metabolism. We find that the loss of p110α, but not p110β, in muscle leads to insulin resistance and reduced muscle mass primarily due to enhanced protein degradation. Interestingly, loss of p110α is also accompanied by increased expression of *PGC1α*, especially the α2 and α3 isoforms, increased mitochondrial biogenesis, altered mitochondrial dynamics, and increased mitochondrial oxidative capacity. Taken together, these data indicate that PI3K and the p110α catalytic subunit play a unique role in regulation of muscle metabolism, being critical for maintenance of muscle mass and normal mitochondrial homeostasis. This bifunctional role of p110α may, in part, explain the association between muscle insulin resistance and mitochondrial dysfunction in diabetes and other insulin-resistant states.

## Results

**Decreased PI3K activity and impaired insulin signaling**. Skeletal muscle expresses both major catalytic subunits of PI3K. RNA-Seq analysis of muscle revealed that *p110α* mRNA was 12-fold higher than *p110β* mRNA when analyzed as the fraction of total reads (Supplementary Fig. 1a), suggesting a dominant role of p110α in this tissue. To more directly address this question, we specifically inactivated the *p110α* or *p110β* genes in skeletal muscle using Cre-lox recombination. The resultant M-p110αKO displayed a 77% and a 83% reduction in *p110α* mRNA and protein levels in pooled hindlimb skeletal muscles, respectively (Supplementary Fig. 1b, c). This percentage approximates the relative contribution of myocytes to other cells in skeletal muscle, such as satellite cells, adipocytes, endothelial cells, nerves, and fibroblasts[13]. Although no good antibodies exist for p110β

protein, there was a similar 79% reduction in *p110β* mRNA in skeletal muscle of M-p110βKO (Supplementary Fig. 1d).

Consistent with the relative abundance of *p110α* versus *p110β* mRNA, insulin-stimulated PI3K activity in anti-IRS-1 and anti-pTyr immunoprecipitates was almost completely lost in muscle extracts of M-p110αKO (Fig. 1a), whereas no significant difference in PI3K activity was detected in M-p110βKO mice (Supplementary Fig. 1e), despite a trend toward reduced pTyr-associated PI3K activity in these mice (IP:pTyr, $p = 0.12$, Student's $t$ test). Similarly, a qualitative comparison of insulin-stimulated PIP3 immunofluorescence—the product of PI3K—showed almost complete loss of PIP3 staining in muscle of M-p110αKO, whereas the PIP3 level in M-p110βKO was similar to the controls (Supplementary Fig. 1f, g). As expected, muscle-specific deletion of p110α impaired insulin signaling with a 42% decrease in phosphorylation of AKT in skeletal muscle (Fig. 1b, c). Somewhat surprisingly, following insulin injection, there was a similar decrease in ERK phosphorylation ($p = 0.04$) (Supplementary Fig. 2a) and a tendency toward reduced 4EBP1 phosphorylation (Supplementary Fig. 2b), suggesting some downstream crosstalk between these pathways. However, there was no significant difference in AKT phosphorylation in M-p110αKO, as compared with controls, following 13 weeks of high-fat diet (HFD) feeding (60% calories from fat) (Supplementary Fig. 2c). ERK phosphorylation was also not different, indicating that insulin resistance induced by HFD was sufficient to abrogate the difference in insulin signaling observed between controls and M-p110αKO on chow diet (CD) (Supplementary Fig. 2c). By comparison, no changes were displayed in the phosphorylation levels of AKT, ERK, GSK3β, or 4EBP1 in M-p110βKO, whether on CD or HFD (Supplementary Fig. 2d, e).

**Increased proteasome activity and autophagy signalings**. Muscle gene expression was assessed using RNA-Seq analysis. Principle component analysis (PCA) revealed distinct gene expression patterns in skeletal muscles of controls and M-p110αKO, both in fed and fasted states (Fig. 1d). Unsupervised KEGG analysis revealed several pathways that were significantly enriched in M-p110αKO, including genes involved in the TCA cycle (Fig. 1e), oxidative phosphorylation (OXPHOS) (Fig. 1f) and lysosome/autophagosome signaling (Fig. 1g). Interestingly, in both TCA cycle and OXPHOS gene sets, discrete subsets of genes were upregulated in fed versus fasted state, suggesting the differences in metabolic adaptation to a state of energy deprivation.

Autophagy is an important pathway that regulates protein degradation in muscle and is regulated by FoxO transcriptional factors[14,15]. Increased autophagy markers were displayed in quadriceps muscle of M-p110αKO, with 53–130% increases in ULK1[555] phosphorylation, a nutrient-dependent initiator of autophagy and 3- to 3.9-fold increases in the total LC3B protein, a marker and mediator of autophagy[16] in both fed and fasted states (Fig. 2a). LC3B levels were not increased with fasting in controls or M-p110αKO, likely due to increased protein degradation, but were increased in both groups of fasted mice after treatment with colchicine to block autophagic flux. In soleus muscle, LC3B protein levels were less upregulated in fed M-p110αKO, and no difference between the groups was seen with fasting (Supplementary Fig. 2f), suggesting that a difference in autophagy is not as prominent in oxidative fibers. Interestingly, p110α expression was also regulated by energy status in quadriceps muscle of controls with a 2.3-fold increase in p110α protein after 24 h of fasting (Fig. 2a). This may explain the different gene expression profile in TA muscle of M-p110αKO between fed and fasted states. Again, an increase in p110α protein upon fasting was not observed in oxidative

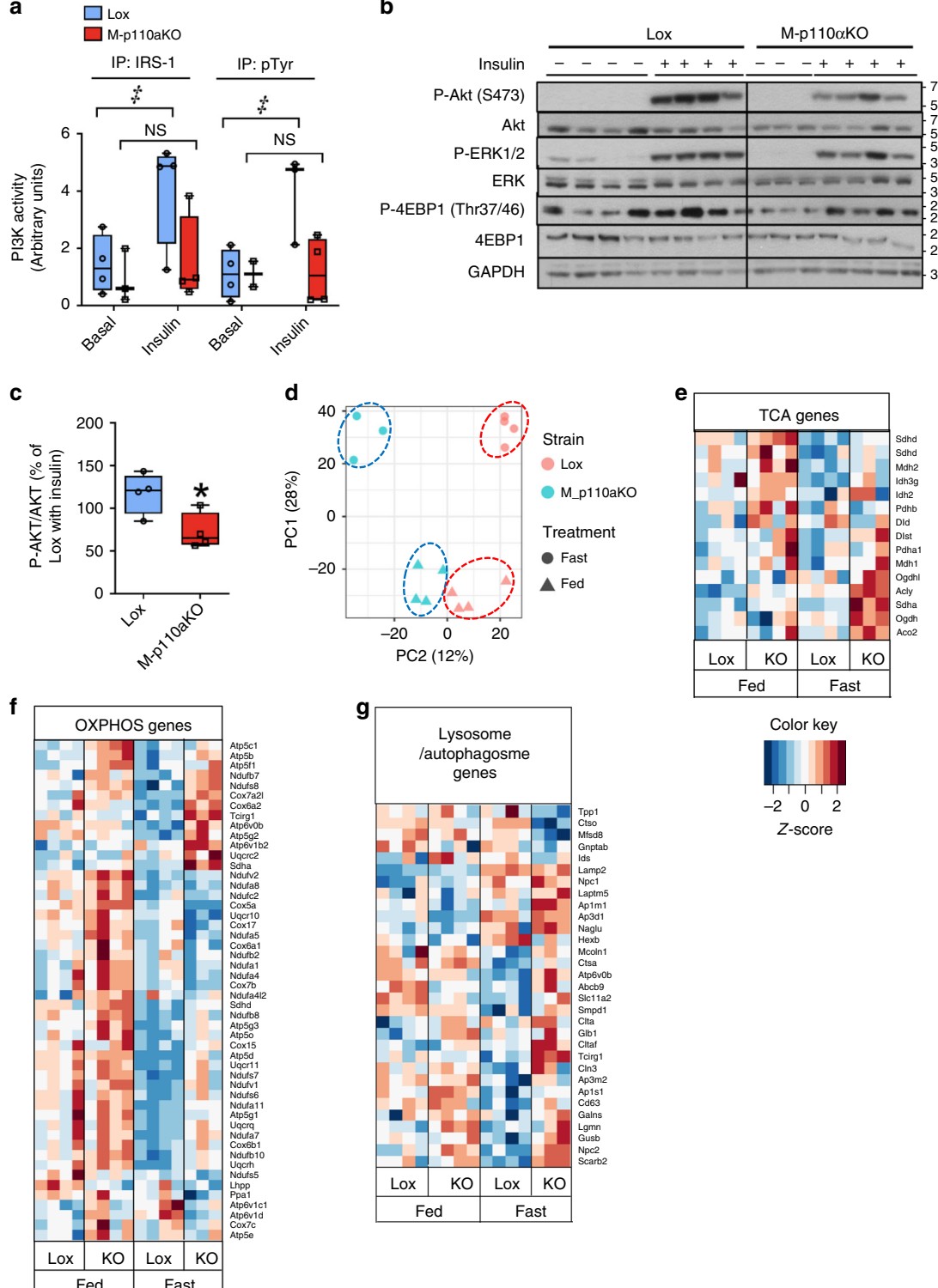

**Fig. 1** Impaired insulin signaling and changes in FA oxidation and lysosome pathways. **a** PI3K activities from extracts immunoprecipitated with anti-IRS-1 or anti-pTyr antibodies in gastrocnemius muscles of M-p110αKO and control mice after insulin injection (5 U of regular insulin via the inferior vena cava) ($n = 4$). **b**, **c** Western blot of insulin signaling (**b**) and densitometric analysis of P-AKT/AKT (**c**) in quadriceps muscle of M-p110αKO and controls that were fasted overnight and treated with saline or insulin intravenously. **d** Principal component analysis (PCA) plot of transcriptomic analysis by RNA-Seq from TA muscles of control and M-p110αKO in fed and 24 -h fasted conditions (ctrl, $n = 3$; KO, $n = 4$). **e**–**g** Heatmap of transcripts involved in KEGG pathways analyzed from the RNA-Seq experiment in panel **d** (ctrl, $n = 3$; KO, $n = 4$). Genes involved in the TCA cycle (**e**), oxidative phosphorylation (**f**), and autophagosome– lysosme pathways (**g**) were listed in the heatmap, and the color intensities indicate Z-score of each sample by gene. All mice were 3-month old. *$P < 0.05$ by Student's $t$ test; ‡$P < 0.05$ by two-way ANOVA. PC1 principle coordinate 1, PC2 principle coordinate 2. Box plots visualize the five-number summary of a data set (minimum, lower quartile, median, upper quartile, and maximum)

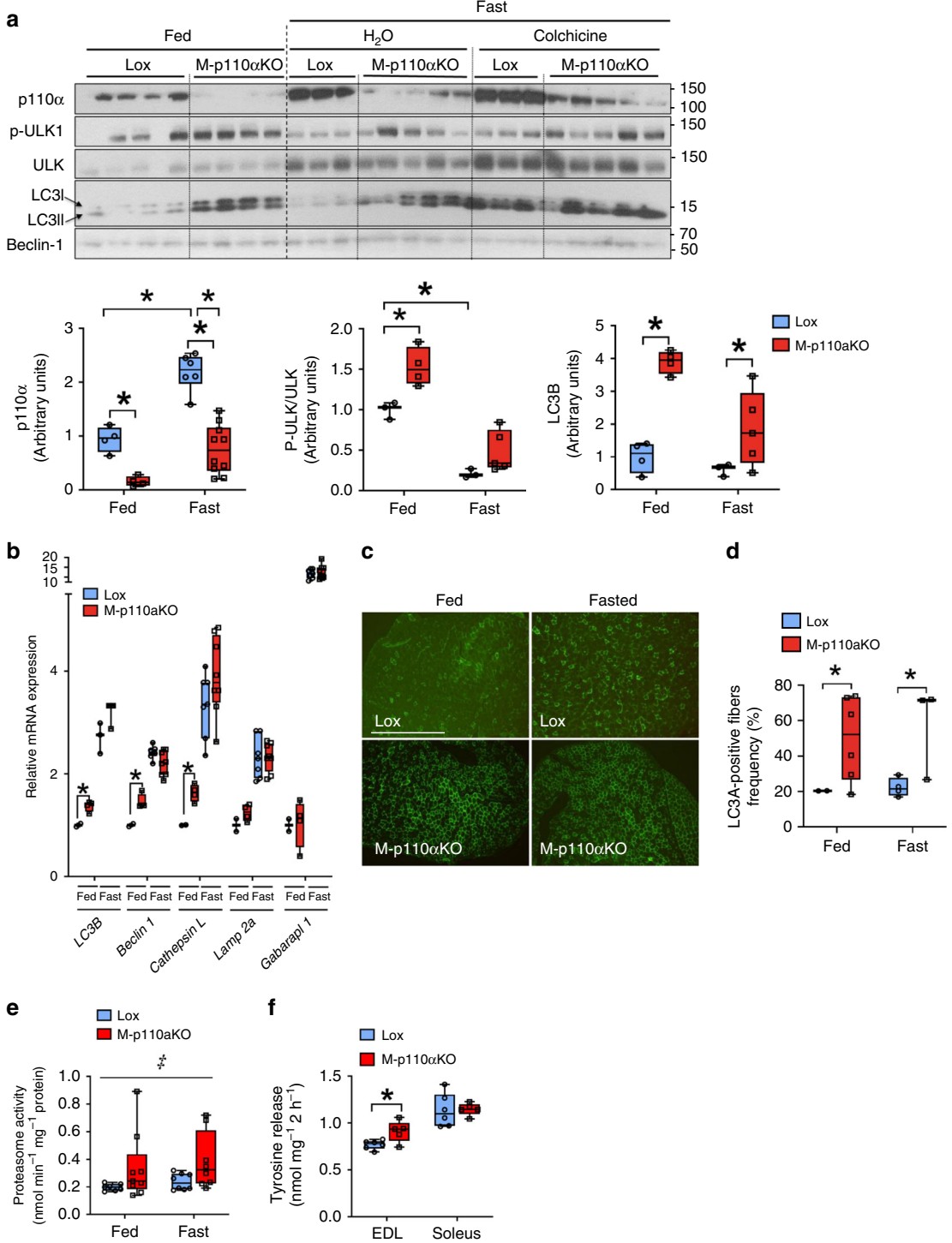

**Fig. 2** Increased markers of autophagy–lysosomal degradation and proteasome activity. **a** Western blot and densitometric analysis of autophagy intermediates in quadriceps muscles of M-p110αKO and controls in fed, fasted, and fasted plus treated with colchicine. **b** mRNA expression of autophagy genes in TA muscles of M-p110αKO and controls ($n = 4$). **c** LC3A immunostaining of quadriceps from M-p110αKO and controls (scale bar = 500 μm). **d** Percentage of LC3A-positive fibers per cross-section of quadriceps muscles in (**c**) ($n = 4$–7). **e** Proteasome activity in gastrocnemius muscles of M-p110αKO and controls using a peptidyl glutamyl-like (LLE) substrate ($n = 8$). **f** Proteolysis rate measured by tyrosine release ex vivo in EDL and soleus muscles isolated from fed control and M-p110αKO at 24 months of age (ctrl, $n = 5$; KO, $n = 6$). Mice for Fig. 2a–e were 3-month old, and mice for Fig. 2f were 24-month old. Mice for Fig. 2a–e were fasted for 24 h or randomly fed. *$P < 0.05$ by Student's $t$ test; ‡$P < 0.05$ by two-way ANOVA. Box plots visualize the five-number summary of the data set (minimum, lower quartile, median, upper quartile, and maximum). The data are mean ± SEM in bar graph

soleus muscle (Supplementary Fig. 2f). In addition to increases in autophagy markers at the protein level, autophagy regulating genes, such as *LC3b*, *beclin1*, and *cathepsin L*, were also increased 40–63% at the mRNA level in glycolytic TA muscle in fed state (all $p < 0.05$, Student's $t$ test) (Fig. 2b). Qualitative immunostaining of muscle sections confirmed increased percent of LC3-positive fibers in M-p110αKO in both fed and fasted conditions (Fig. 2c, d).

Increased proteasome activity was also observed in muscle extracts of M-p110αKO in both fed (72%) and fasted (62%) conditions using peptidyl glutamyl-like (LLE) substrates (Fig. 2e). This occurred with no change in mRNA expression of genes that encode proteasome subunit proteins (Supplementary Fig. 3a), suggesting increased proteasome activity may be induced by a post-transcriptional or conformational change in the protein[17]. Proteolysis demonstrated a 19% increase in tyrosine release from extensor digitorum longus (EDL), but not soleus muscle, in M-p110αKO at 24 months of age ($p < 0.05$, Student's $t$ test) (Fig. 2f), indicating muscle-specific increases in proteolysis at an age generally accompanied with muscle wasting[18]. These changes were not observed, however, in M-p110αKO at 5 months of age, when there was minimal loss of muscle mass (Supplementary Fig. 3b). Together these data indicate that both the proteasome pathway and the autophagy pathway were upregulated in M-p110αKO.

Forkhead box O (FoxO) transcription factors are downstream targets of the PI3K-AKT axis and regulate skeletal muscle mass through both proteasome and autophagy pathways[15,19,20]. In the absence of AKT-mediated phosphorylation, FoxO proteins are translocated from the cytosol to the nucleus where they upregulate expression of atrophy-related genes. In spite of decreased AKT activation/phosphorylation, there was no difference in FoxO nuclear translocation in M-p110αKO and controls (Supplementary Fig. 3c).

In summary, loss of p110α, but not p110β, leads to decreased insulin signaling as documented by decreased PI3K activity, reduced PIP3 levels, and diminished AKT phosphorylation. This leads to increased autophagy and proteasome degradation in glycolytic and mixed muscle fibers, resulting in increased muscle atrophy with aging. These results are in agreement with previous studies from our lab reporting that a loss of insulin signaling in muscle in muscle-specific IR and IGF1R knockout mice leads to increased muscle atrophy[10].

**Larger mitochondria due to altered mitochondrial dynamics.** Ultrastructure of the mitochondria was assessed in transverse sections of EDL and soleus muscles by electron microscopy (EM). In glycolytic EDL muscle, M-p110αKO displayed an 83% increase of average mitochondrial area compared with controls, whereas there was no difference in the more oxidative soleus muscle (Fig. 3a, b). This occurred with no change in the mitochondrial number in either EDL or soleus muscles in M-p110αKO (Supplementary Fig. 4a). In control mice, average mitochondrial area was ~90% greater in soleus than EDL muscle, reflecting more slow, oxidative fibers in the soleus compared with EDL.

To directly assess the effect of muscle-specific p110α deletion on mitochondrial morphology in vivo, we transfected a mito-GFP plasmid that specifically targets the mitochondrial matrix[21] into quadriceps muscles of living mice using a gene gun delivery system[22]. Quantitation using confocal microscopy showed that the total mitochondrial area (Fig. 3c) and the average size of a single mitochondrion (Fig. 3d) were increased by 2- to 2.5-fold in M-p110αKO mice. The increase in the total mitochondrial area was further accentuated by a decrease in the muscle fiber size as intravital imaging also confirmed a 17% decrease of the muscle fiber area, as quantified from intramyofibrillar images in M-p110αKO mice (Supplementary Fig. 4b).

Mitochondrial fusion and fission regulate mitochondrial morphology and play important roles in mitochondrial quality control[23]. Optic atrophy 1 (OPA1) and mitofusin 2 (Mfn2) are proteins that drive mitochondrial inner- and outer membrane fusion[24], and their protein levels were increased by 36% and 38% in TA muscle of fed M-p110αKO, respectively (all $p < 0.05$,

Student's $t$ test) (Fig. 3e). Dynamin-related protein 1 (Drp1) is a driver of mitochondrial fission and is translocated from the cytosol to mitochondrial membranes upon dephosphorylation of Ser637[25]. Previous studies have shown that during fasting, mitochondrial fission is inhibited to sustain the ATP demand during nutrient deprivation[26]. In fed state, M-p110αKO had a 50% increase in Drp1 phosphorylation at Ser637 compared with controls ($p < 0.05$, Student's $t$ test) (Fig. 3e), creating a decreased drive for mitochondrial fission in these mice. Thus, loss of p110α in muscle shifts mitochondrial dynamic toward an increased fusion and decreased fission, resulting in larger and more elongated mitochondria in M-p110αKO.

**Increased mitochondrial biogenesis in M-p110αKO.** The levels of mitochondrial- and nuclear-encoded proteins involved in OXPHOS were assessed using muscle extracts. In fed state, loss of muscle p110α led to a 69% increase in protein levels of NDUFB8 in Complex I, and a 280% increase in succinate dehydrogenase subunit B (SDHB) from Complex II (Fig. 4a, b). Upon 24 h of fasting, M-p110αKO displayed further increases of mitochondrial protein expression, including SDHB (about 200%), ubiquinol cytochrome c reductase core protein 2 (UQCRC2) of Complex III (86%), and ATP5A of Complex V (50%) (all $p < 0.05$, Student's $t$ test). These changes in mitochondrial proteins were accompanied by increase in mitochondrial respiratory enzyme activities in muscle of M-p110αKO, with a 14% increase in Complex IV activity ($p < 0.05$, Student's $t$ test) (Fig. 4c) and tendencies of increased activities of Complexes I to III and II to III in the mitochondria of fasted M-p110αKO (Supplementary Fig. 4c, d). Citrate synthase (CS) catalyzes the first step in the TCA cycle, and is a marker for the intact mitochondria[27]. Loss of p110α in muscle led to a 45% increase in CS protein expression (Fig. 4d) and a parallel increase in enzyme activity in M-p110αKO in fed and fast states (Fig. 4e).

PGC1α1 is a transcription factor that controls oxidative metabolism in muscle and other tissues[28]. Four transcripts of $PGC1\alpha$ (1, 2, 3, and 4) are produced from $PGC1\alpha$ gene and regulate the expression of different downstream genes[29]. PGC1α1 functions as a major regulator of mitochondrial biogenesis, whereas PGC1α4 is mostly known to promote muscle hypertrophy[30]. Little is known about the functions of PGC1α2 and PGC1α3. Muscle-specific deletion of p110α led to increased mRNA expression of all $PGC1\alpha$ isoforms, with 45–55% increases of $PGC1\alpha1$ and $PGC1\alpha4$ and 70–80% increases in expression of $PGC1\alpha2$ and $PGC1\alpha3$ (all $p < 0.05$, Student's $t$ test) (Fig. 4f). Upon fasting, mRNA levels of $PGC1\alpha1$ and 4 were unchanged, while $PGC1\alpha2$ and 3 were increased by 3.3- and 2.8-fold in controls and further increased in M-p110αKO by three- to fivefold.

**Increased mitochondrial oxidative capacity in M-p110αKO.** Our findings of increased mitochondrial mass and biogenesis prompted us to measure mitochondrial respiratory capacity and ATP content. Consistent with the increased mitochondrial mass and increased mitochondrial function, ATP levels were increased by 26% in quadriceps muscle of M-p110αKO compared with controls (Fig. 5a). This was accompanied by a 96% increase in fatty acid oxidation in the mitochondria isolated from hindlimb muscle of M-p110αKO (Fig. 5b). In addition, there was a 93% increase in oxygen consumption rate (OCR) in presence of ADP (state 3) with pyruvate/malate as substrates in mitochondria isolated from skeletal muscle of M-p110αKO, indicating enhanced oxidative capacity (Fig. 5c). These changes occurred with no changes in mRNA expression of genes involved in fatty acid oxidation in skeletal muscle of M-p110αKO (Supplementary Fig. 4e).

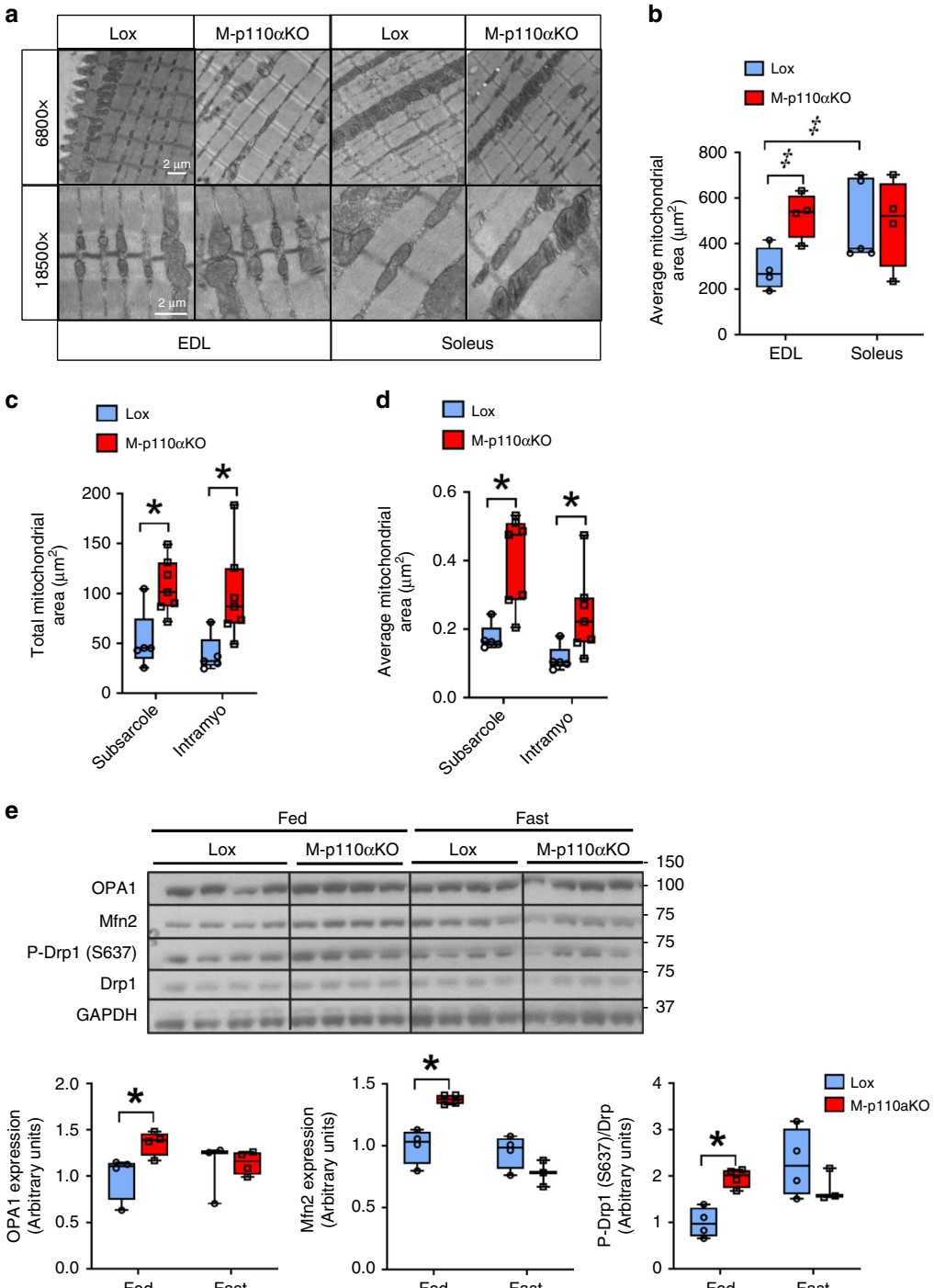

**Fig. 3** Larger mitochondria associated with altered mitochondrial dynamics. **a** Ultrastructure of the mitochondria in transverse planes of EDL and soleus muscles from M-p110αKO and controls viewed by EM. **b** Quantification of the average mitochondrial size from EDL and soleus muscles in panel **a** ($n = 4$). **c**, **d** Quantification of the total mitochondria area per muscle fiber view–sight (**c**) and the average mitochondrial area (**d**) from M-p110αKO and controls transfected with Mito-GFP in panel **c** (ctrl, $n = 3$; KO, $n = 4$ with four to ten fibers per mouse). **e** Western blot and densitometric analysis of proteins involved in mitochondrial dynamics in TA muscles of M-p110αKO and controls fasted for 24 h or randomly fed. All mice were 4- to 5-month old. *$P < 0.05$ by Student's t test; ‡$P < 0.05$ by two-way ANOVA. Electron microscopy EM, subsarcole subsarcolemma, intramyo intramyofibrillar. Box plots visualize the five-number summary of the data set (minimum, lower quartile, median, upper quartile, and maximum)

NADH is a major product of the TCA cycle, and can be visualized in muscle of living mice by its autofluorescence[31]. In muscle of control mice, NADH autofluorescence was 77% lower in intramyofibrillar regions compared with those in subsarcolemmal regions (Fig. 5d), consistent with other studies showing lower mitochondrial function in the intramyofibrillar mitochondria[32]. Compared with controls, NADH levels were increased in both intramyofibrillar (~8-fold) and subsarcolemmal (~13.6-fold) locations in M-p110αKO, consistent with higher TCA cycle activity. This increased mitochondrial metabolic capacity was associated with increased free radical production in M-p110αKO with 55–60% increases in combined reactive

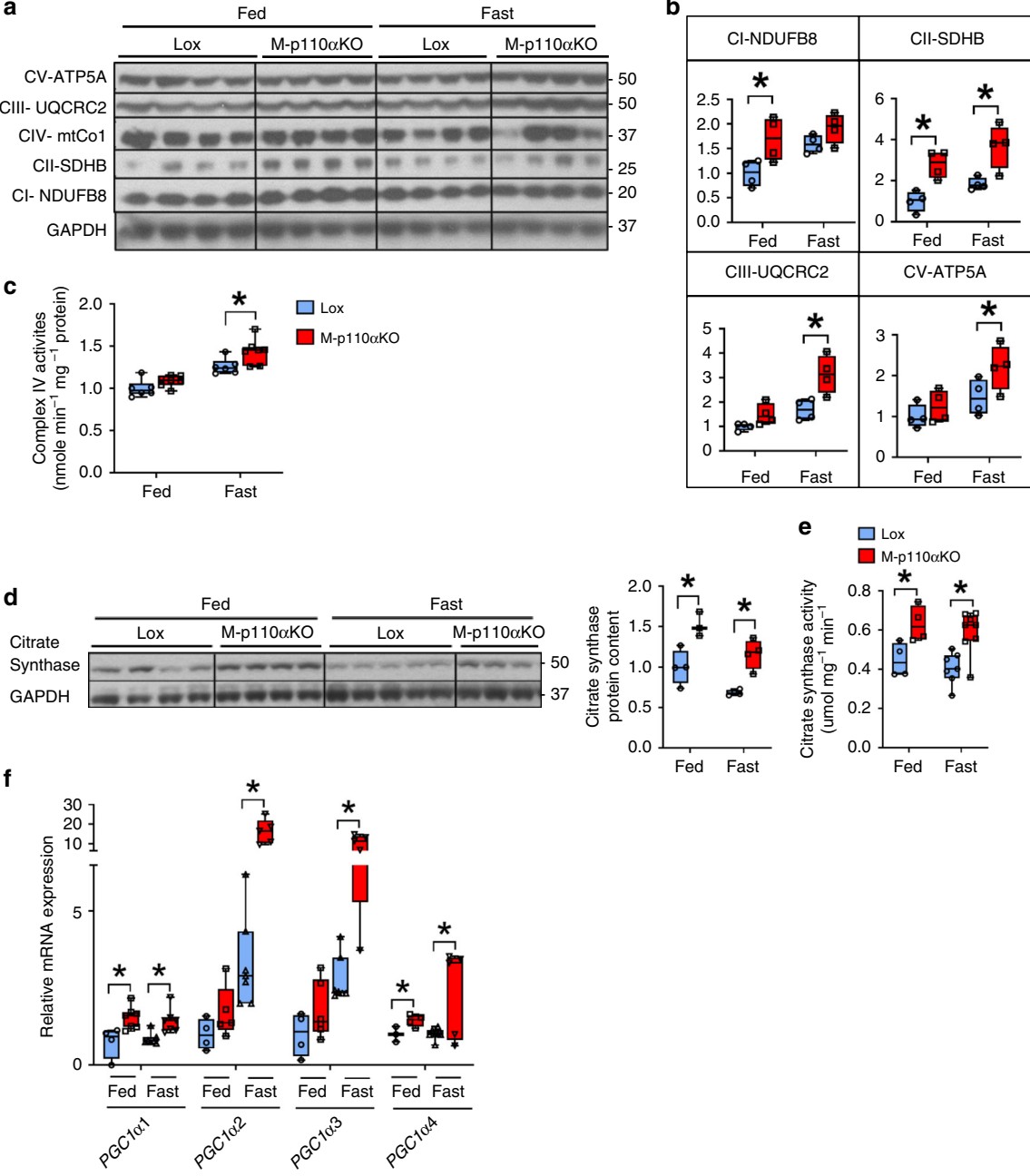

**Fig. 4** Increased mitochondrial biogenesis and enhanced expressions of *PGC1α* isoforms. **a**, **b** Western blot (**a**) and densitometric analysis (**b**) of proteins from the OXPHOS pathway in TA muscles of M-p110αKO and controls fasted for 24 h or randomly fed. **c** Complex IV activity in mitochondria isolated from skeletal muscle of M-p110αKO and controls in fed or 24 h fasted conditions (ctrl, $n = 6$; KO, $n = 7$). **d** Citrate synthase protein content from TA muscles of M-p110αKO and controls fasted for 24 h or randomly fed as measured by western blot analysis. **e** Citrate synthase activity from gastrocnemius of M-p110αKO and control mice fasted for 24 h or randomly fed (ctrl, $n = 4$; KO, $n = 9$). **f** mRNA expression of *PGC1α* isoforms in TA muscles of M-p110αKO and control mice fasted for 24 h or randomly fed (ctrl, $n = 4$; KO, $n = 9$). All mice were 3- to 4-month old. *$P < 0.05$ by Student's $t$ test. Box plots visualize the five-number summary of the data set (minimum, lower quartile, median, upper quartile, and maximum). NDUFB8 ubiquinone oxidoreductase subunit B8, SDHB succinate dehydrogenase subunit B, UQCRC2 ubiquinol cytochrome c reductase core protein 2, ATP5A ATP synthase α-subunit

oxygen and reactive nitrogen species (ROS/RNS) levels in muscle of M-p110αKO in both fed and fasted states (Fig. 5e). This was confirmed by in vivo fluorescence microscopy, when muscle fibers of living mice were transfected in situ using a gene gun delivery approach with a pMitoTimer reporter plasmid, which targets to the mitochondria and irreversibly shifts the fluorescence spectrum from green to red in the presence of ROS[33,34]. Quantitation showed a 45–50% increase in red to green fluorescence ratio in M-p110αKO in both the subsarcolemmal

(Fig. 5f, g) and intramyofibrillar (Supplementary Fig. 4f, g) mitochondria, indicating higher free radical release in muscle of these mice. This shift in MitoTimer fluorescence in M-p110αKO was abolished when the muscle was co-transfected with a constitutively active mutant of p110α (PIK3CA) E545K (Fig. 5g; Supplementary Fig. 4g). Together, these data show elevated mitochondrial oxidative capacity and increased ATP content, accompanied by increased oxidative stress in muscle following p110α deletion.

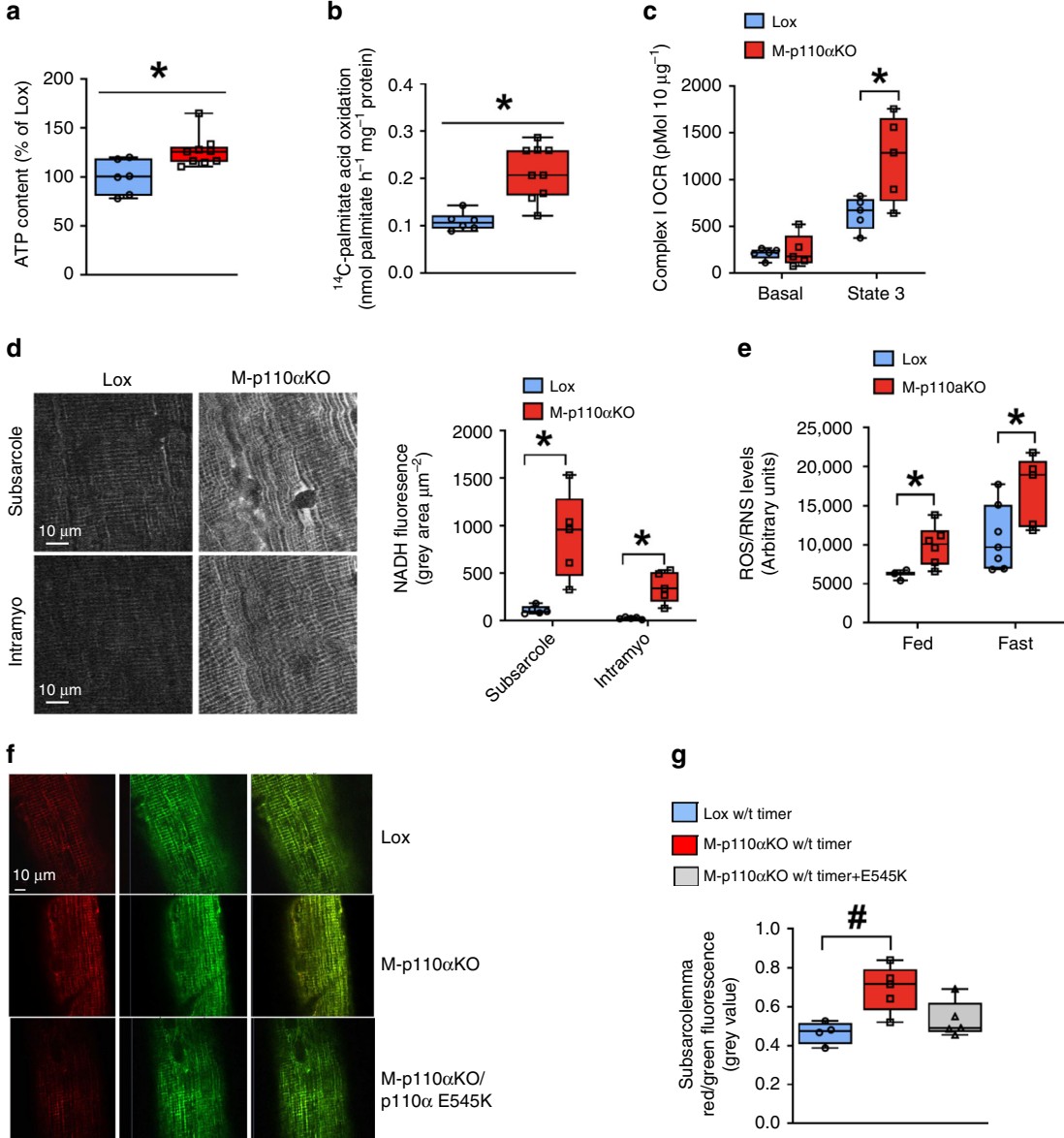

**Fig. 5** Increased mitochondrial oxidative capacity in skeletal muscle of M-p110αKO. **a** ATP content in quadriceps muscles of M-p110αKO and control mice in fed state (ctrl, $n = 8$; KO, $n = 9$). **b** Fatty acid oxidation was determined by measuring $^{14}CO_2$ production in the mitochondria isolated from hindlimb of overnight fasted mice using $^{14}C$-palmitate acid as a substrate (ctrl, $n = 7$; KO, $n = 9$). **c** Basal and ADP-dependent (State 3) respiratory rates of the isolated mitochondria from hindlimb of fed M-p110αKO and control mice were measured using a Seahorse X24 Flux Analyzer in the presence of a Complex I substrate (pyruvate/malate) (ctrl, $n = 5$; KO, $n = 6$). **d** Intravital multiphoton images of mitochondria-specific NADH autofluorescence in quadriceps muscles of M-p110αKO and control mice after overnight fasting (scale bar = 10 μm). **e** Levels of total free radicals composed of reactive oxygen species (ROS) and reactive nitrogen species (RNS) measured in gastrocnemius muscles of M-p110αKO and control mice (ctrl, $n = 5$; KO, $n = 9$). **f** pMitoTimer plasmid was in situ transfected into quadriceps muscle fibers of living mice using a gene gun delivery approach and visualized 5 days later by the confocal microscope (upper and middle set of images). A different cohort of M-p110αKO mice was co-transfected with constitutively active mutant p110α (PIK3CA) E545K into quadriceps of these mice (bottom set of images). **g** Red/green fluorescence in subsarcolemmal position of quadriceps muscles in (**f**) ($n = 4$ with 3–4 fibers per mouse). All mice were 4- to 5-month old. *$P < 0.05$ by Student's $t$ test; #$P < 0.05$ by one-way ANOVA. Subsarco subsarcolemma, intramyo intramyofibrillar, OCR, oxygen consumption rate, Timer pMitoTimer, E545K p110α (PIK3CA) E545K. Box plots visualize the five-number summary of the data set (minimum, lower quartile, median, upper quartile, and maximum). The data are mean ± SEM

**Hyperinsulinemia and insulin tolerance on HFD and aging.** At 3 months of age, M-p110αKO displayed a 60% increase in serum insulin levels in fed state (Fig. 6a), but intraperitoneal glucose-tolerance tests (GTT) revealed no difference between M-p110αKO and controls (Fig. 6b). Likewise, insulin-tolerance tests (ITT) were not statistically different, although there was a tendency toward impaired insulin sensitivity in M-p110αKO (Fig. 6c). After 13 weeks on 60% HFD, both M-p110αKO and

controls had 1.5- to 3-fold increases in serum insulin, but no difference was found between the groups (Supplementary Fig. 5a). GTT and ITT were also not different between M-p110αKO and controls on HFD (Supplementary Fig. 5b, c). In general, increased insulin resistance occurs with aging[35]; however, 16-month-old M-p110αKO displayed no difference in whole-body glucose home-ostasis compared with controls as measured by intraperitoneal GTT and ITT (Supplementary Fig. 5d, e). Since the whole-body

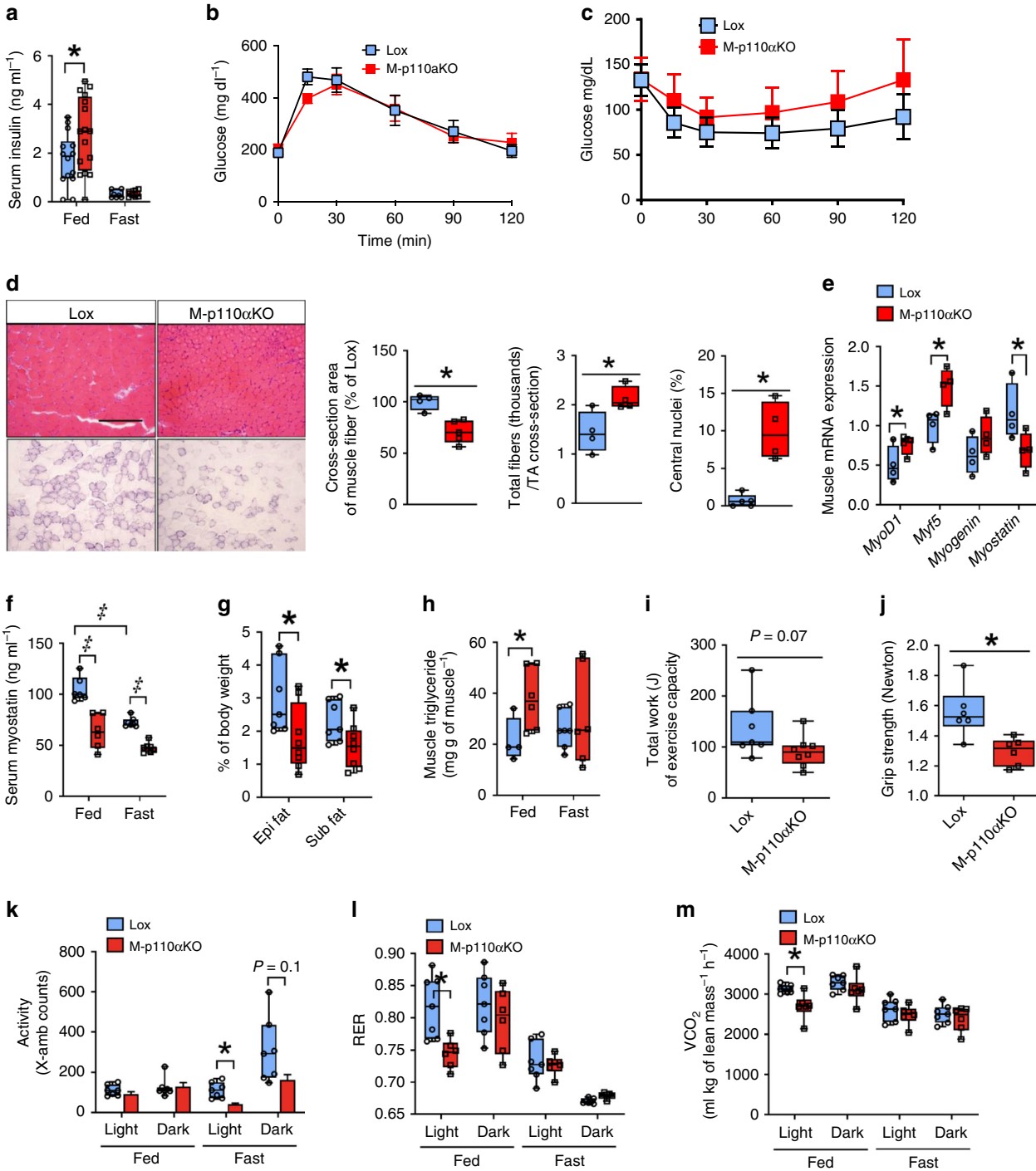

**Fig. 6** Hyperinsulinemia, abnormal muscle function, and decreased RER. **a** Serum insulin levels from M-p110αKO and control mice fasted for 24 h or randomly fed at 3 months of age (ctrl, $n = 5$; KO, $n = 7$). **b, c** Intraperitoneal glucose tolerance test (GTT) and insulin-tolerance test (ITT) (**c**) performed in M-p110αKO and control mice at 3 months of age on chow diet (CD) (ctrl, $n = 5$; KO, $n = 8$). **d** H&E stain in quadriceps and succinate dehydrogenase (SDH) stain in TA muscle cross-sections of M-p110αKO and control mice at 3 months of age (scale bar = 200 μm), and quantification of the muscle fiber cross-section area from H&E stain, total fiber numbers per area of TA muscle cross-section from SDH stain, and percentage of quadriceps myofibers with central nuclei from H&E stain ($n = 5$). **e** mRNA expression of myogenic factors in TA muscles from 3-month-old M-p110αKO and control mice ($n = 5$). **f** Serum myostatin levels from 3-month-old M-p110αKO and control mice fasted for 24 h or randomly fed (ctrl, $n = 5$; KO, $n = 7$). **g** Percentage of epididymal fat and subcutaneous fat weights per body weight of M-p110αKO and control mice at 3 months of age ($n = 8$). **h** Triglyceride levels in gastrocnemius muscles of 3-month-old M-p110αKO and control mice fasted for 24 h or randomly fed (ctrl, $n = 4$; KO, $n = 9$). **i** The total work of M-p110αKO and control mice from acute treadmill test at 16 months of age ($n = 8$). **j** Forelimb grip strength from 16-month-old M-p110αKO and control mice ($n = 8$). **k–m** Spontaneous activity (**k**), respiratory exchange ratio (RER) (**l**), and $VCO_2$ (**m**) in 24-month-old M-p110αKO and control mice during fed and fast cycles over 48 hrs in CLAMS metabolic cages (ctrl, $n = 5$; KO, $n = 6$). *$P < 0.05$ by Student's $t$ test; ‡$P < 0.05$ by two-way ANOVA. Light blue bars represent p110α-floxed mice; red bars represent M-p110αKO mice. TA tibialis anterior, Gastroc gastrocnemius, Quad quadriceps, EDL extensor digitorum longus. Box plots visualize the five-number summary of the data set (minimum, lower quartile, median, upper quartile, and maximum). The data are mean ± SEM

glucose tolerance and insulin sensitivity were not different between M-p110αKO and controls, we assessed in vivo glucose uptake specifically in muscle and adipose tissue utilizing tritiated glucose. Interestingly, despite impaired insulin signaling in M-p110αKO, there was no significant difference in basal or insulin-stimulated glucose uptake in TA and soleus muscle in controls and M-p110αKO (Supplementary Fig. 5f). Likewise, as expected, M-p110βKO as compared with controls at 3 months of age had no difference in GTT and ITT, both on CD and HFD, although all mice showed higher glucose levels throughout GTT and ITT on HFD (Supplementary Fig. 5g, h).

**Decreased muscle mass, fiber size, and increased myogenesis.**
At 3 months of age, M-p110αKO displayed a 15% decrease of body weight compared with controls (Supplementary Fig. 6a). This was due, in large part, to 12–24% reductions of mass in most muscles (all $p < 0.05$, Student's $t$ test) (Supplementary Fig. 6b), except soleus muscle, whose weight was not different between the groups. By contrast, no difference was observed in the body or muscle weight of M-p110βKO at 3 months of age compared with controls (Supplementary Fig. 6c, d). The reduced muscle mass in M-p110αKO was due to a 30–40% decrease of the muscle fiber cross-sectional area as quantitated using hematoxylin and eosin (H&E) and succinate dehydrogenase (SDH) stained sections (Fig. 6d; Supplementary Fig. 6e). There was also a 48% increase in the muscle fiber number in cross-sections of quadriceps muscles, and a ten-fold increase in myofibers with central nuclei in M-p110αKO (10% vs. 1% in controls) (Fig. 6d), indicating increased myogenesis in these mice[36]. SDH staining of TA muscles showed a preponderance of glycolytic fibers with no difference between controls and M-p110αKO (Supplementary Fig. 6f, g). Consistent with increased myogenesis, there was increased expression of *MyoD1* (49%), *Myf5* (50%), and *myogenin* (39%) in TA muscle of 3-month-old M-p110αKO (all $p < 0.05$, Student's $t$ test) (Fig. 6e).

Expression of *myostatin*, a TGFβ family member that has an inhibitory role in muscle growth[37], on the other hand, was decreased by 44% in muscle of M-p110αKO (Fig. 6e). The decrease in *myostatin* and increase in *MyoD1* were also observed even after 24 -h fasting (Supplementary Fig. 6h), when protein catabolism of muscle is increased[38]. Previously we have generated mice with a heterozygous knockin mutation changing Arg 649 to Trp in the gene for the p85α PI3K regulatory subunit. This results in significantly decreased insulin signaling[39]. This was also associated with decreased myostatin expression and a tendency for increased *MyoD1* mRNA, thus providing further evidence that the impaired PI3K signaling results in decreased myostatin expression (Supplementary Fig. 6i).

The decrease in *myostatin* mRNA in M-p110αKO was associated with a parallel reduction of serum myostatin levels in fed and fasted states (Fig. 6f). Reduction in circulating myostatin has been associated with increased muscle mass and decreased fat mass[40]. In M-p110αKO, there was a 35% decrease in weight of the epididymal fat pad, and a 28% decrease in weight of the subcutaneous fat pad compared with controls ($p < 0.05$, Student's $t$ test) (Fig. 6g). The decreased adipose tissue mass was associated with increased expression of genes involved in lipolysis, including ~70% increases in mRNA of adipocyte triglyceride lipase (ATGL) and hormone-sensitive lipase (HSL) (Supplementary Fig. 6j). Likewise, mRNA levels of genes that regulate β-oxidation were increased in adipose tissue of M-p110αKO, including peroxisome proliferator-activated receptor-α (PPARα) (145%), PGC1α1 (341%), very-long chain acyl-CoA dehydrogenase (VLCAD) (48%), and long-chain acyl-CoA dehydrogenase (LCAD) (62%) ($p < 0.05$, Student's $t$ test) (Supplementary Fig. 6k).

The combined reduction in muscle and fat mass was associated with a 76% increase in triglyceride accumulation in muscle of M-p110αKO in fed state, and a margin 34% increase in the fasted state (Fig. 6h). This was associated with a 60% increase in expression of fatty acid transport protein 1 (FATP1), the major transporter for fatty acid uptake in muscle of M-p110αKO (Supplementary Fig. 7a)[41]. These changes occurred with no changes in mRNA expression of genes involved in lipolysis in muscle of M-p110αKO (Supplementary Fig. 7b).

**Abnormal muscle function and decreased RER in M-p110αKO.**
While no functional defect in endurance or strength was present in M-p110αKO at 3 months of age (Supplementary Fig. 7c, d), by 16 months of age M-p110αKO exhibited a 33% decrease of exercise capacity on a treadmill test ($p = 0.07$) (Fig. 6i) and a 16% decrease of muscle grip strength ($p < 0.05$, Student's $t$ test) (Fig. 6j, Student's $t$ test). Likewise, in CLAMS metabolic cage assessment at 3 months of age, M-p110αKO displayed similar food intake, activity, $O_2$ consumption, $CO_2$ production, and respiratory exchange ratio (RER), versus controls in both fed and fasted conditions (Supplementary Fig. 7e–i). However, by 24 months of age, fasted M-p110αKO exhibited 47–63% decreases in spontaneous activity compared with controls, as assessed in CLAMS metabolic cages (Fig. 6k). Food intake, adjusted for body weight, was not different between M-p110αKO and controls at 24 months of age (Supplementary Fig. 7j). RER was decreased by 9% in M-p110αKO during the light cycle under fed conditions (0.74 ± 0.01 vs. 0.81 ± 0.023, $p < 0.05$, Student's $t$ test) (Fig. 6l), due to a 13% of decrease in $CO_2$ production in M-p110αKO (Fig. 6m), indicating a shift of energy utilization from carbohydrate toward fat. Oxygen consumption was not different between M-p110αKO and controls in either fed or fasted conditions (Supplementary Fig. 7k). Together, these data suggest a shift of energy utilization from carbohydrate consumption toward fatty acid oxidation in aged M-p110αKO mice, accompanied by decreased spontaneous activity and exercise capacity.

## Discussion

PI3K and its product PIP3 are the central signaling nodes in insulin regulation of metabolism[1,12,42,43]. In this study, we have investigated the roles of two major isoforms of the PI3K catalytic subunit, p110α and p110β, in regulation of metabolism in skeletal muscle. Similar to the liver, we find that a loss of p110α, but not p110β, in muscle leads to impaired insulin signaling, as evidenced by diminished phosphorylation of its downstream target AKT. This difference between a loss of p110α and p110β can be explained, at least in part, by different gene expression, as *p110α* mRNA is 12-fold higher than *p110β* in muscle. In addition, it is possible that p110α and p110β have different intrinsic functions as suggested by our data and previous studies[2,3].

Although deletion of p110α, but not p110β, results in impaired muscle insulin signaling and hyperinsulinemia, these mice had normal whole-body glucose tolerance and insulin sensitivity. Furthermore, glucose uptake in skeletal muscle is also not impaired in M-p110αKO as measured in vivo by radioactively labeled glucose. Likewise, glucose tolerance and insulin sensitivity are not different even after 13 weeks on HFD or with aging, suggesting that in vivo, the defect in insulin signaling produced by a loss of p110α in muscle only is not sufficient to impair glucose tolerance.

Normal whole-body glucose tolerance in M-p110αKO is consistent with our findings of normal glucose tolerance in mice with muscle-specific deletion of insulin receptor[44] or insulin and IGF-1 receptors[10]. The latter has a complete lack of insulin signaling, but an increase in non-insulin-mediated glucose uptake,

secondary to decreased TBC1D1 resulting in increased Glut4 membrane localization[10]. Whether this is true in muscle of M-p110αKO remains to be determined. Along the same line, muscle-specific deletion of the regulatory p85α subunit of PI3K, which destabilizes and reduces p110α protein, does not result in glucose intolerance[43]. By contrast, work from our lab and others have shown that mice with muscle-specific knockout of Glut4, which is responsible for both insulin-mediated and exercise-induced glucose uptake, develop markedly impaired glucose tolerance[45], indicating that muscle glucose uptake, when completely impaired, is important in whole body glucose homeostasis.

Despite a lack of obvious defects in glucose metabolism, a loss of p110α does induce fasting-like phenotype in muscle characterized by increased mitochondrial function and fatty acid oxidation. This is evident by increased levels of electron transport chain complex proteins and enhanced expression and enzymatic activity of citrate synthase in muscle of M-p110αKO. There is also an eightfold increase in mitochondrial NADH levels, an increase in cellular ATP levels, and increased $C^{14}$-labeled palmitate oxidation in the isolated mitochondria. However, this shift toward increased fatty acid oxidation comes at a cost of higher ROS levels. Loss of p110α is also associated with increased proteasome and autophagy–lysosomal muscle degradation, reduced muscle mass, and smaller muscle fibers, especially in glycolytic muscles, whereas predominantly oxidative soleus muscle appears to adapt to higher fatty acid oxidation. These findings are in line with some clinical studies showing increased mitochondrial activity and fatty acid oxidation with the development of muscle insulin resistance in T2DM and obesity[6].

Increased mitochondrial activity in M-p110αKO is likely due to increased expression of *PGC1α* isoforms. Indeed, as a major regulator of mitochondrial biogenesis, mice overexpressing PGC1α have more newly formed muscle fibers and an increase in mitochondrial mass[46], both features of M-p110αKO. Mitochondrial fusion, a process that promotes ATP production, is increased in M-p110αKO, as a result of increased expression of regulators of mitochondrial fusion such as Mfn2. This process may also be regulated by the increased *PGC1α* in M-p110αKO[47]. Other studies have shown a strong correlation between intramuscular lipid storage and increased PGC1α expression[48]. This is also true in M-p110αKO mouse muscle, which has increased triglyceride accumulation. Indeed, many studies have shown that several weeks of HFD treatment in rodents induces an increase in mitochondrial biogenesis in skeletal muscle[8]. This elevated fatty acid influx could induce an adaptive increase in mitochondrial biogenesis and function in muscle, as a way to achieve energy homeostasis. However, the magnitude of these changes in mitochondria appears to be insufficient to overcome enhanced lipid availability, therefore ectopic lipid accumulation is observed in M-p110αKO. Another interesting aspect of these changes is that the increase in *PGCα1* occurs primarily in two of the less studied isoforms, *PGC1α2* and *PGC1α3*, and how these differ from the effects of the better characterized *PGC1α1* and *PGC1α4* remains to be determined.

In any case, the loss of p110α results in increased mitochondrial biogenesis and function. This appears to be a compensatory mechanism secondary to a decrease in insulin signaling and glycolytic capacity. This is evident by higher levels of ROS—a by-product of mitochondrial respiration—in muscle of M-p110αKO. Previous studies have shown that ROS produced by the mitochondria can stabilize LC3, the key factor in autophagosome biogenesis, and therefore is essential for stress-induced autophagy[49,50]. Indeed, high levels of ROS have been shown to increase proteolysis and decrease protein synthesis, contributing to the pathogenesis of muscle atrophy[51]. In line with this, increased autophagy signaling is displayed in M-p110αKO along with

decreased skeletal muscle mass. Interestingly, mitochondrial size, LC3-mediated autophagy, increased proteolyis, and decreased muscle mass were only evident in glycolytic and mixed muscle fibers, but not in oxidative muscle such as soleus, which is able to adapt better to fatty acid oxidation.

In M-p110αKO, there is also decreased circulating myostatin. This results in increases in myoblast proliferation, numbers of muscle fibers per muscle, and numbers of immature muscle fibers with central nuclei[52]. This increased myogenesis is a compensatory mechanism secondary to increased muscle protein degradation, such that muscle mass is reduced by only 10–15% in M-p110αKO, despite the relatively large changes in each of the individual processes controlling maintenance of muscle mass. Interestingly, a decrease in PI3K signaling in M-p110αKO associated with a decrease in muscle expression of myostatin is also evident in our mice with dominant-negative mutation in the p85α subunit of PI3K, suggesting a direct role of PI3K in muscle homeostasis. This finding raises concern that prolonged use of PI3K inhibitors in cancer patients, especially those targeted against p110α[53], may add to muscle atrophy already present in many of these individuals.

Whereas decreased myostatin levels were not sufficient to overcome reduced muscle size in M-p110αKO, it could account for increased lipolysis and reduced whole-body fat mass, contributing to relatively mild metabolic impairment. Indeed, previous studies have shown that inhibition of myostatin action or myostatin deficiency results in increased muscle mass accompanied by decreased fat mass[54]. M-p110αKO have 40–50% decreases of *myostatin* expression in muscle and serum myostatin. This may contribute to increased adipose tissue lipolysis through increased expression of *ATGL* and *HSL* in adipose tissue[55]. In M-110αKO, there is also an increase in *FATP1* in muscle, which could lead to increased lipid uptake by muscle[56], decreased adipose tissue mass, and accumulation of triglycerides in muscle of M-p110αKO, a phenotype found in some models of muscle atrophy[57]. Myostatin inhibition is also one possible driver for increased *PGC1α* expression in M-p110αKO. Indeed, previous studies have shown PGC1α expression is induced in myostatin inhibition models[58], and in line with this, M-p110αKO display increased mRNA expression of *PGC1α* in both skeletal muscle and white adipose tissue, suggesting a systemic regulation of *PGC1α* gene expression.

In summary, our study demonstrates that p110α is the dominant catalytic isoform of PI3K in muscle and plays a unique role in metabolic regulation. p110α is critical for the maintenance of muscle mass. It also plays an important and unexpected role in the regulation of mitochondrial homeostasis. Deletion of p110α in muscle leads to significant increase in proteasome activity and autophagy, resulting in reduction of muscle mass, strength, and endurance, especially with age. Interestingly, loss of p110α in skeletal muscle causes an increase, rather than decrease, in mitochondrial metabolism, as evidenced by increases in mitochondrial biogenesis, mitochondrial fusion, oxidative capacity, and levels of ROS. This is likely due to increased expression of *PGC1α* isoforms found in M-p110αKO. Thus, changes at the level of p110α may help explain differences in muscle mitochondrial function in the context of diabetes. These data shed light on how p110α regulates metabolic functions in skeletal muscle and add a new layer of understanding the relationship between mitochondrial function and insulin sensitivity.

## Methods

**Animal care and use**. All studies were performed in male mice on *C57BL/6J* background. Muscle-specific p110α or p110β knockout mice were generated by crossing mice carrying the Cre recombinase gene driven by the human alpha-skeletal actin (HSA) promoter (Jackson Laboratories Stock Number: 006149) with

mice carrying either floxed p110α or p110β alleles, in which exon 1 of p110α or exon 2 of p110β was flanked with loxP sites[2,3,59]. All animal studies were approved by the Institutional Animal Care and Use Committee (IACUC) at the Joslin Diabetes Center, and were in accordance with the National Institutes of Health guidelines. See Supplementary Methods for more information.

**RNA-Seq processing and data analysis**. RNA-Seq analysis was performed as previously reported[60]. Library for RNA-Seq was prepared using the NEBNext mRNA Sample Prep Master Mix kit (NEB), and HTG EdgeSeq mRNA sequence analysis was performed by BioPolymers Facility at Harvard Medical School. The total RNA was extracted from tissues using Trizol and was reverse-transcribed using a high-capacity complementary DNA reverse transcription kit (Applied Biosystems) according to the manufacturer's instructions. Quantitative real-time PCR was performed in 5 μl of the resulting cDNA after a tenfold dilution in the presence of the SYBR Green PCR Master Mix (Applied Biosystems) and 300 nM primers. PCR reactions were run in duplicate in the ABI Prism 7700 Sequence Detection System with primers as detailed in Supplementary Table 1, and Ct values were normalized to *GAPDH* gene levels. KEGG enrichment analysis was performed using a hypergeometric distribution test. Filtering of gene expression and statistics were performed using MATLAB 2016b.

**Proteolysis assay**. Proteolysis was measured as previously described[61]. Briefly, EDL and soleus muscles were isolated from M-p110αKO and controls, and pre-incubated in 1 ml of KRB buffer (117 mM NaCl, 4.7 mM KCl, 2.5 mM CaCl$_2$, 1.2 mM KH$_2$PO$_4$, 1.2 mM MgSO$_4$, 24.6 mM NaHCO$_3$, and 5 mM glucose) for 30 min before transferring to fresh 1 ml of KRB containing 0.5 mM cycloheximide to inhibit protein synthesis. Incubation buffer was collected, and tyrosine release was measured[62].

**Proteasome measurement**. Proteasome activity was measured from gastrocnemius muscles of M-p110αKO and controls fasted as previously described[15]. Briefly, muscle fractions containing proteasome were isolated from homogenized muscle lysates after centrifugation. A peptidyl glutamyl-like (LLE) substrate (Z-Leu-Leu-Glu-7-amido-4-methylcoumarin) (Sigma-Aldrich, C0483) was added into assay buffer, and fluorescence was monitored (360-nm excitation, 460-nm emission) every 3 min for 1.5 hrs at 37 °C. Enzyme activity of 26S proteasome was determined as the change in fluorescence during the linear phase of the reaction and compared against a standard curve of 7-amido-4-methylcoumarin (Sigma-Aldrich, A9891).

**Mitochondrial respiratory enzyme activities**. Mitochondrial assays were performed according to established methods[63]. The NADH cytochrome $c$ reductase (NCCR) and succinate cytochrome $c$ reductase (SCCR) activities were measured by following the reduction of exogenous oxidized cytochrome $c$. An aliquot of 10 μg of isolated mitochondria was preincubated with the assay buffer (1.5 mM KCN, 50 mM K$_2$HPO$_4$, pH 7.4) containing β-NADH or succinate at 37 °C for 15 min. After addition of cytochrome $c$ to the mixture, the change in the absorbance at 550 nm was recorded on a spectrophotometer. Cytochrome $c$ oxidase (CCO) activity was determined by following the oxidation of exogenous reduced cytochrome $c$. An aliquot of 5 μg of isolated mitochondria was preincubated in the assay buffer (5 mM K$_2$HPO$_4$, pH 7.4) at 37 °C for 10 min. After addition of ferrocytochrome $c$ to the assay mixture, the change in absorbance at 550 nm was recorded on a spectrophotometer.

**EM and morphometric analysis**. After perfusion with PBS, soleus and EDL muscles were dissected from mice and fixed using 2.5% (wt/vol) glutaraldehyde in 0.1 M phosphate buffer for 4 hrs at room temperature before being placed at 4 °C for storage. Sample preparation for EM was performed by EM Core Facility of Harvard Medical School[64]. After being placed in ascending concentrations of ethanol, samples were incubated in 1:1 mixture of propylene oxide and Araldite 502 epoxy resin for several hours before being placed into a vacuum desiccator overnight. This was followed by embedding samples in Araldite 502 epoxy resin using BEEM capsules (Ted Pella Inc.) and a 60 °C curing oven for 48 hrs. Sections were trimmed and adhered to glass slides using heat, stained with a 1% methylene blue, and prepared for EM. Morphological analysis such as mitochondrial size and area was obtained using Metamorph software (v 6.1, Universal imaging)[34]. At least four sections were measured per animal in a double-blind manner.

**Plasmid transfection and intravital microscopy**. For in vivo imaging, mice were anesthetized and mounted on the microscope stage on their side in dental cement in the position to expose the tissue to be imaged[22]. Images were collected with a 63 ×, 1.2 NA Zeiss C-Apochromat objective on a Zeiss-LSM-710 microscope. Mito-GFP was excited using the 488-nm laser line. Mito-EGFP emission was collected at 505–576 nm. MitoTimer was excited using the 488 (green) and 561-nm laser line (red). MitoTimer emissions were collected at 505–540 nm (green channel) and 590–640 nm (red channel). NADH was excited and collected as previously described[64].

**Fatty acid oxidation by isolated mitochondria**. Mitochondria were isolated and purified from hindlimb muscle of M-p110α and controls as previous described[13], and fatty acid oxidation was measured by the conversion of [1–$^{14}$C]-palmitate acid into CO$_2$ in isolated mitochondria[65]. Briefly, isolated mitochondria were placed in a buffer containing BSA-conjugated [1–$^{14}$C]-palmitate acid for 1 h at 37 °C. CO$_2$ was released from buffer by addition of 70% perchloric acid, and was captured in 3 M NaOH that had been added to a filter paper on top of the vial. Fatty acid oxidation was calculated based on the level of $^{14}$CO$_2$ trapped in the filter paper by scintillation counting.

**In vivo glucose uptake**. Glucose uptake was measured as described[66]. In brief, glucose uptake into tissue was measured by intravenous injection of 0.33 μCi of H$^3$-labeled glucose per gram of body weight dissolved in either saline or 20% dextrose, administered via the retro-orbital sinus. After 25 min, disintegrations per min were measured in selected tissue and in blood at 0, 5, 10, and 15 min.

**Oxygen consumption rate analysis**. Mitochondria were isolated and purified from hindlimb muscle, then diluted in the Mitochondrial Assay Solution supplied with 10 mM pyruvate/2 mM malate as previously described[13]. Oxygen consumption rate (OCR) was measured using a Seahorse Bioscience XF24 analyzer in the absence (basal) or presence (state 3) of 400 μM adenosine diphosphate (ADP). Protein concentrations were measured using the Bradford method (Bio-Rad).

**Western blot analysis**. Powdered muscle tissue was homogenized in RIPA buffer (Millipore, 20–188) supplemented with phosphatase inhibitor and protease inhibitor cocktail (Sigma-Aldrich). Lysates were subjected to SDS-PAGE and blotted using following antibodies: p110α (CS4249, 1:1000), phospho-Akt (S473) (CS9271, 1:2000), phospho-ERK (CS9101, 1:2000), phospho-GSK3β (CS9336, 1:2000), phospho-4EBP1 (CS2855, 1:1000), phospho-ULK (S555) (CS5869, 1:1000), phospho-Drp1 (S637) (CS4867, 1:1000), phospho-FoxO1/3 (CS9464, 1:500), ERK (CS9102, 1:2000), GSK3β (CS9315, 1:2000), 4EBP1(CS9644, 1:500), LC3 (CS2775, 1:1000), Beclin-1(CS3495, 1:1000), Drp1 (CS8570, 1:1000) and FoxO1 (CS2880, 1:500) from Cell Signaling, OPA1 (SC393296, 1:1000), GAPDH from Santa Cruz (SC25778, 1:5000), and citrate synthase (Ab96600, 1:1000), Mfn2 (Ab56889, 1:1000), and OXOPHOS (Ab110413, 1:1000) proteins from Abcam. Uncropped blots are available in Supplementary Fig. 8.

**Statistical analysis**. The data are expressed as mean ± SEM. Differences were analyzed using an unpaired Student's $t$ test, One-way ANOVA or two-way ANOVA as appropriate. One-way ANOVA was followed by the Tukey's multiple comparison post hoc test, and two-way ANOVA was performed using Bonferroni's multiple comparison post test. Statistical calculations were performed using the GraphPad Prism software (GraphPad, San Diego, CA). A probability value of < 0.05 was considered significantly different.

**Reporting summary**. Further information on research design is available in the Nature Research Reporting Summary linked to this article.

## Data availability
All relevant data are included in the article and Supplementary Information. The RNA-seq data generated in this study are available at NCBI GEO database with the accession number GS E124394. Additional data generated during and/or analyzed during this study are available from the corresponding author on reasonable request.

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

## Acknowledgements

This work was supported by NIH grants RO1DK055545 and RO1DK033201 (to C.R.K.). M.E.L. was funded by a NIH Training Grant T32 DK007260, and B.T.O. was funded by a K08 training award from the NIDDK of the NIH (K08DK100543). The Joslin Diabetes Center DRC core facility was used for part of this work (P30 DK36836).

## Author contributions

M.E.L. designed the study, researched data, and wrote the paper. H.P.M.M.L. and B.T.O. researched the data, helped design experiments, and helped write the paper. C.W., B.B., W.C., M.S., R.T. and M.F.H. researched data and helped design experiments. S.S., helped design experiments, generated additional data and revised the manuscript. C.R.K. designed the study, helped write the paper, and oversaw the research.

## Additional information

**Competing interests:** The authors declare no competing interests.

