## [Peer Review File · Nature Communications]

Reviewers' Comments:

Reviewer #1:

Remarks to the Author:

In the submitted manuscript, Li and colleagues characterize skeletal muscle specific p110a and p110b knockout mice. The stated premise for the studies is to advance the understanding of the role of skeletal muscle insulin resistance and mitochondrial dysfunction in the pathogenesis of type 2 diabetes. The authors find that p110a is the dominant PI3K catalytic subunit in skeletal muscle. The authors additionally report that deletion of p110a is associated with reduced muscle mass and increased rates of autophagy and proteasomal activity, despite decreases in plasma myostatin levels and increased expression of myogenic factors in muscle. They further observe increased mitochondrial mass in p110a MKO mice, presumably resulting from increased PGC1 isoform expression. Finally, despite increased muscle triglyceride levels and impaired insulin signaling, p110a KO mice display normal glucose tolerance.

The major strengths of the manuscript are the use of novel animal models and a number of rigorous techniques to evaluate insulin signaling, mitochondrial function, autophagy, proteasomal activity and whole-body energy balance and glucose tolerance. Another strength is that the manuscript has the potential to influence thinking regarding the cause and effect nature of insulin resistance and mitochondrial dysfunction/compensation during the pathogenesis of type 2 diabetes. The main weakness is the descriptive nature of the manuscript and lack of mechanistic insight regarding most of the novel findings. There are also a few instances where the interpretation of the data is overstated and causality is ascribed to associative observations (noted in detail below). Also of concern is that some of the major claims are largely predictable, specifically that impaired insulin signaling is associated with reduced muscle mass in association with increased proteasomal activity, and literature speaking to these known phenomena do not receive sufficient attention (more detail below). In this same vein, there is no attempt to connect the loss of p110a and PI3K activity to increased PGC1 expression in a mechanistic fashion, either through experimentation or discussion of the literature. Finally, given the stated premise of the manuscript, insufficient attention is given to characterizing changes in skeletal muscle glucose transport and insulin resistance in vivo.

Specific comments:

A major claim of the paper is that p110a, and not p110b, is the major PI3K catalytic subunit in skeletal muscle. PI3K activity and signaling data from RC fed to support this claim are convincing, but there are some subtleties that deserve discussion. First, the fact that pAKT levels following insulin treatment are not statistically different in HFD-fed lox vs M-p110a mice should be described as such. When describing insulin signaling results on page 5 line 4, the authors should note no significant difference in pAKT in M-p110a mice and give the P-value, if they wish to comment about a trend towards reduced levels. Insulin resistance in the lox mice may be blunting the differences observed between groups on RC. While there was no significant difference in IRS1-associated PI3K activity in the M-p110b KO mice, there does appear to be a reduction compared with controls and it would be more balanced to describe this supplemental data as such. Finally, if the results in Supplemental data 1f and 1g are going to be described as similar to the PI3K activity data, then they should be analyzed, ie mean \pm sem and a statistical comparison, in the same fashion. Otherwise it should be noted that this is a qualitative comparison.

The statement that increased proteasomal and autophagosomal pathway activities are due to decreased FoxO1 phosphorylation and activity in the results section is an overstatement. This should be described as an association and limited only to the phospho FoxO1 data as there is no direct measure of FoxO1 activity presented. Also, the statement is somewhat contradictory to the observation that there were no differences in proteasome-related gene expression and the authors' own statement that "increased proteasomal activity may be induced by a post-transcriptional or conformational change in the protein." Alternatively, measuring transcriptional readouts of FoxO1

activity that mediate muscle atrophy via the ubiquitin-proteasomal pathway – MuRF1 and atrogin-1 – would strengthen conclusions related to FoxO1. There are a number of references that describe how loss in insulin receptor signaling results in FoxO1 mediated skeletal muscle atrophy via the ubiquitin-proteasomal pathway and they should be discussed. Finally, phospho FoxO1 data is shown in the fasted state and may be more meaningful in fed-state. Nuclear localization by microscopy or cell fractionation would be more convincing than phospho levels alone.

Given what appear to be fiber type specific effects of p110a deletion, particularly the mitochondrial morphology data, it is critical to include a more thorough characterization of the p110a deletion in different muscle depots/fiber types (EDL, soleus, TA, gastrocnemius). Throughout the manuscript different depots are used without rationale for why this was done or why it may not matter and this should be discussed. For example, KO and signaling data in Fig. 1 are from mixed fiber type sources (gastroc and quad), while RNA-seq data in this figure are from a primarily fast/glycolytic depot (tibialis ant). Follow up studies in Fig. 2 exploring the differences in autophagy gene expression from RNA-seq in figure 1 use mixed fiber quadriceps and gastroc for protein measures and proteasome activity, while gene expression is measured from fast/glycolytic tibialis anterior. The observation that age-dependent muscle proteolysis (2f), autophagy gene expression (2b) and mitochondrial area (3a) are only different in fast/glycolytic depots (TA and EDL) would argue for focusing here, and may explain some of the inconsistencies in other figures, such as the lack of effect of fasting on ULK1 phosphorylation and the LC3II/I ratio in mixed fiber tissues in 2a and 2c. The fact that fasting does not appear to increase pULK1 or LC3II/I ratio compared with fed samples (control or KO) should be discussed.

GTTs were performed after an overnight fast and this may account for the lack of difference as fasting of this duration is insulin sensitizing in the mouse. Relative hyperinsulinemia in the fed-state suggests some degree of insulin resistance. The conclusion that there is no difference in glucose homeostasis would be more convincing if additional studies were performed. A short term fast prior to GTT is suggested and may reveal impaired glucose tolerance, as suggested by the fed plasma insulin levels and would be more physiologically relevant. Measuring plasma insulin during the GTT minimally at the 0 and 30 min time points may reveal insulin resistance/compensatory insulin secretion. Ideally a hyperinsulinemic euglycemic clamp could be performed to resolve whether p110a deletion in muscle impairs insulin-stimulated glucose disposal. While initial characterization of the muscle specific insulin receptor KO (MIRKO) mice demonstrated no change in glucose tolerance, subsequent clamp studies in these mice demonstrated impaired whole-body insulin sensitivity due to impaired skeletal muscle glucose uptake (Kim JK, JCI, 2000. PMID 10862794). These data should be discussed in the context of this model, as well.

Another modest oversight in the manuscript is the lack of discussion of work from Lewis Cantley's group looking at the effects of p85 subunit deletion and subsequent effects on glucose tolerance and muscle mass. Deletion of p85 isoforms is reported to reduce p110 protein levels as well, resulting in similar reductions in muscle mass, but contradictory impaired glucose tolerance compared to data reported here (Luo J, Cell Metab, 2006. PMID 16679293). A more complete comparison of the p110a MKO mice to MIRKO and other skeletal muscle Akt or p85 null models is warranted in the discussion, specifically with regards to changes in muscle mass, mitochondrial mass and glucose intolerance/insulin resistance.

Muscle weight should be expressed as a percentage of body weight, as is done for fat mass, or per tibia length. This will distinguish whether muscle mass is specifically reduced or if these are smaller mice in general, which the reduced body weight suggests.

The statement in the results that increased skeletal muscle triglyceride is due to increased FATP1 expression is an overstatement. A more balanced interpretation is that increased triglyceride was associated with increased FATP1. An alternative interpretation is that reduced insulin stimulated glucose transport due to p110a deletion produced a starvation like effect in muscle where increased reliance on fatty acids was a feature, such that FATP1 expression was induced.

Food intake should be normalized to lean mass for consistency with all other metabolic cage data reported instead of per mouse.

Minor:

Typo on page 4, third line from the bottom where "complete loss of PI3K staining" should read "complete loss of PIP3 staining."

Data in S2a is described in the results as quantification of pAkt, but shows p4EBP1 data.

The units in Fig. 6h seem incorrect – mg triglyceride per gram tissue or protein are typical, not a concentration of triglyceride (mg/dl) as is used here.

Reviewer #2:

Remarks to the Author:

In this paper the authors investigate the role of the catalytic subunit of PI3K, p110a in skeletal muscle. In line with previously reported observations in liver and adipose tissue, the authors show that in muscle only the loss of p110a, and not p110b, perturbs the PI3K-AKT cascade and therefore results in impaired insulin signaling. The phenotypes have been analyzed after different nutritional challenges (fasting or HFD) and comparing effects observed in young and old mice. Although obesity and diabetes are usually associated with mitochondrial dysfunction, the authors show in this work that increased mitochondrial respiratory capacity, associated to an induction of PGC1a expression in mice lacking p110a in skeletal muscle. Phenotypically, reduced AKT pathway activation is associated with decreased muscle mass and function.

The phenotyping of these mice at the organismal level is complete and thorough and allows the authors to clearly demonstrate the mild nature of muscle dysfunction. Unfortunately, the authors have a tendency to oversell the phenotypic relevance of this mouse model as it clearly does not reproduce human diabetes: rather it is a mild model of starvation-like phenotype. It should be made clearer in the discussion that the increase in mitochondrial respiration might be a putative compensatory mechanism that, together with increased activity of other organs, prevents the development of the disease. Neither of these hypotheses have been experimentally tested and thus the study, while interesting, remains largely descriptive. I feel that the manuscript warrants further revision prior to publication.

Major comments:

The quality of the immunoblots in Fig. S2 makes it difficult to draw clear conclusions and is somewhat problematic with respect to P-GSK and P-4EBP where some bands are rounded, and P-FoxO that seems overexposed.

Fig. 1b and S2c should be shown in one panel in order to conclude about the effect of HFD.

Fig. S2c, the text refers to ERK but the figure show GSK, which one is correct?

Fig. 2a, no effects of fasting in control mice were observed, which is surprising? What is the explanation? It would be helpful for the authors should provide other autophagy markers and the phosphorylation status of AKT and its targets.

Fig. 2c, the image does not appear to be representative of the differences that are reported by quantification. Fibers are clearly different and it is difficult to understand the scoring of LC3 positivity. Moreover, these data appear incongruent with the protein levels illustrated by immunoblot (Fig. 2a) and the transcript levels (Fig. 2b) where the differences between KO and control mice are mainly in fed condition

The mitochondrial morphology and structure revealed by electron microscopy show clear differences Fig. 3c however it is unclear whether these are cause or consequence. Could it be that fibre disruption (see Figure 2c) leads to downstream mitochondrial perturbations? Differential transfection capacity of mutant and wild type muscle fibers could explain the observed differences. The mitochondrial morphology images provided in Figure 3c are unconvincing and it is unclear how one can define mitochondrial area using these data. Moreover, I call into question the relevance of mitochondrial dynamics in this model and doubt that the mild alterations in steady state levels of the fusion and fission proteins reported in Fig. 3f are sufficient to alter mitochondrial morphology directly. Moreover, mitochondrial morphology in muscle fibers appear similar between mutant and wild type in Fig 5F. How can the data in this figure and Fig 3F this be reconciled?

DRP1 activity is depending on the balance in the phosphorylation of two different serine residues, authors should provide the status of P-DRP1 (S616). If increased fusion is at play, attempts to reduce fusion capacity either by downregulating OPA1 or both Mitofusins (both of which are reportedly upregulated) would be predicted to rescue the phenotypes. Indeed, muscle-specific deletion of OPA1 has been demonstrated to affect insulin signaling and metabolism and thus it would be interesting to know how the authors data fit into this paradigm.

In Fig. 6e and S6g, the data do not support the conclusion that after fasting the increase in myogenetic markers are further exaggerated.

In Fig. 6f, the authors discover reduced circulating myostatin levels and argue that it is associated with increased muscle mass. However, this mouse model exhibits reduced muscle mass, accompanied by increased % of central nucleated fibers and increased expression of myogenetic markers and fibers number, but also increased TG accumulation and decreased fiber area. A proliferation assay would help to better clarify this phenotype

Minor comments:

- _ 3rd paragraph of the introduction: decreased instead of decreased
- _ Reference to Fig. S1c is missing
- _ to be able to compare the effect obtained in the p110a KO compare to the lack of effects in the p110b KO, Fig. S1f and S1g should be set to the same level of signal intensity
- _ Fig. S2e, authors should confirm that the increase P-AKT in p110b KO mice is not statistically significant under CD
- _ Fig. S3b, the age of mice in figure legend does not correspond to the one specified in the text
- _ The reference to Fig. 3a and 3b should go after next sentence
- _ Fig. 3f, authors should confirm that the increase P-DRP1 (S637) in Lox mice is not statistically significant upon fasting
- _ Fig. S4e (or Fig. S7a), given the results obtained analyzing the expression of PGC1a, it would be interesting in these screening to include PPARa.
- _ Fig. 5d, the levels in fed mice should be shown as reference
- _ Fig. 5e, no correspondence between text, graph and figure legend, is it ratio ROS/RNS?
- _ Fig. 5f and S4f, figure legend should be corrected with the proper muscle type
- _ Fig. S5a, the HFD treatment duration indicated in figure legend does not correspond to the one specified in the text
- _ Fig. S6g, correct myogenetic in the text
- _ Fig. 6h, the age of the mice used for the TG measurement should be specified
- _ Fig. S6, it would be interesting to include in the analysis TG levels and lipogenesis status in liver

Reviewer #3:

Remarks to the Author:

The study from the Kahn lab relates to the role of the catalytic PI3K isoforms in skeletal muscle.

By various studies in animals they decipher a positive role of the p110alpha (rather than beta) isoform in metabolic action and muscle mass (incl. inhibition of protein degradation). Yet, the muscle of the transgenic p110 alpha KO animals displays enhanced mitochondria capacity/function. A phenotype proposed to be mediated by elevated PGC1a expression and hypothesized to result in enhanced NADH content/production and free radical release. The author, conclude that the p110 alpha isoform is a key controller in the insulin induced regulation of metabolism, muscle mass and mitochondria homeostasis.

The amount of data provided in this investigation is overwhelming and in full support the ideas put forward by the authors. Also, the study / analyses and methodology applied are of high quality. So in many ways this is a trustful study providing new interesting insights.

Having this huge capacity and skills (Kahn lab), I would have liked to see the authors pushing the ideas to the limits at which the model proposed do not hold as tight or at all. It would be beneficial for the whole society if these limitations become to display – also in journals like Nat Comm. So let us hear the critical voice in this study – telling the reader where – in the view of the authors – the limitations are.

Below I have put forward some thoughts that could be considered:

If myostatin is key to the phenotype (as proposed in the discussion) – it might be of interest to explore whether PI3K is a key player in myostatin expression/release or receptor expression?

The idea that PGC1a-> increase mito-function -> increase ROS production -> increase autophagy, - is interesting and well supported by the data. I recall that the antioxidant capacity has also been positively associated to enhance PGC1a expression. Was this evident here as well? How well is the decreased muscle function related to the loss of muscle mass. Clearly the endurance is unrelated to the mito-function/capacity.

Somewhat ignored in the paper is the interesting observation that the phenotype observed in white muscle is not copied in the red- type II rich – muscle. Why is this? Here is an interesting observation that might point to some interesting mechanisms. Also, 50% of human muscle mass is composed of type I fibers – so the observation in soleus may also relate to human-translational aspects.

The authors relates to possible association between insulin action and mitochondria function.

Although congenital insulin receptor signaling defect has been shown to lead (associate) to mitochondria dysfunction in man (e.g. Kristensen et al: PMID: 24510228 DOI: 10.1007/s00125-014-3187-y), the causality of these parameters is highly controversial in man mainly because of poor matching between cohorts (e.g. T2DM and controls) – in particular- to the main determinant of mitochondria biogenesis (physical activity). The mouse model presented here may provide some further insight to this association (although opposite to the above human findings) if used in a more detailed temporal manner. Alternatively, an inducible KO model may be of value also. Thus,- for me -deciphering the temporal aspect of development for the phenotype could add huge to understanding of any causality. I do not ask the authors to add in a new model – but some thoughts about this would be of value for the paper.

It is interesting that the signaling defect only lead to moderate insulin resistance. The Kahn Lab has previously shown that physical activity is a major confounder for the phenotype of the MIRKO mouse model (PMID: 10545524 PMID: PMC409827 DOI: 10.1172/JC17961). I wonder if the p110alpha KO phenotype is also affected by the lesser level of physical activity.

I believe that several papers have shown that PI3K inhibition by wortmannin and the LY compound affects both the Akt- and the Erk signaling partway in muscle – and thus I do not understand the authors are surprised by this page 5 (top)

Response to reviewers' comments

RE: NCOMMS-17-32216.

Reviewer 1

We would like to thank the reviewer for the generally positive review and helpful suggestion. We especially appreciate the comments that prompted us to more thoroughly investigate the difference in phenotype observed in different muscle fiber types.

Reviewer #1 (Remarks to the Author):

In the submitted manuscript, Li and colleagues characterize skeletal muscle specific p110a and p110b knockout mice. The stated premise for the studies is to advance the understanding of the role of skeletal muscle insulin resistance and mitochondrial dysfunction in the pathogenesis of type 2 diabetes. The authors find that p110a is the dominant PI3K catalytic subunit in skeletal muscle. The authors additionally report that deletion of p110a is associated with reduced muscle mass and increased rates of autophagy and proteasomal activity, despite decreases in plasma myostatin levels and increased expression of myogenic factors in muscle. They further observe increased mitochondrial mass in p110a MKO mice, presumably resulting from increased PGC1 isoform expression. Finally, despite increased muscle triglyceride levels and impaired insulin signaling, p110a KO mice display normal glucose tolerance.

The major strengths of the manuscript are the use of novel animal models and a number of rigorous techniques to evaluate insulin signaling, mitochondrial function, autophagy, proteasomal activity and whole-body energy balance and glucose tolerance. Another strength is that the manuscript has the potential to influence thinking regarding the cause and effect nature of insulin resistance and mitochondrial dysfunction/compensation during the pathogenesis of type 2 diabetes. The main weakness is the descriptive nature of the manuscript and lack of mechanistic insight regarding most of the novel findings. There are also a few instances where the interpretation of the data is overstated and causality is ascribed to associative observations (noted in detail below). Also of concern is that some of the major claims are largely predictable, specifically that impaired insulin signaling is associated with reduced muscle mass in association with increased proteasomal activity, and literature speaking to these known phenomena do not receive sufficient attention (more detail below). In this same vein, there is no attempt to connect the loss of p110a and PI3K activity to increased PGC1 expression in a mechanistic fashion, either through experimentation or discussion of the literature. Finally, given the stated premise of the manuscript, insufficient attention is given to characterizing changes in skeletal muscle glucose transport and insulin resistance in vivo.

Specific Comments:

A major claim of the paper is that p110a, and not p110b, is the major PI3K catalytic subunit in skeletal muscle. PI3K activity and signaling data from RC fed to support this claim are convincing, but there are some subtleties that deserve discussion. First, the fact that pAKT levels following insulin treatment are not statistically different in HFD-fed lox vs M-p110a mice should be described as such. When describing insulin signaling results on page 5 line 4, the authors should note no significant difference in pAKT in M-p110a mice and give the P-value, if they wish to comment about a trend towards reduced levels

We have now indicated that while there was a trend toward reduced levels of pAkt, there was no statistically significant difference in pAkt between control and M-p110 α KO mice following 13 weeks on HFD. ERK phosphorylation was also not different, indicating that insulin resistance induced by HFD in lox mice is enough to abrogate the difference in insulin signaling observed between lox and M-p110 α KO mice on chow diet. (Results, page 5, lines 6-10).

Insulin resistance in the lox mice may be blunting the differences observed between groups on RC. While there was no significant difference in IRS1-associated PI3K activity in the M-p110b KO mice, there does appear to be a reduction compared with controls and it would be more balanced to describe this supplemental data as such

While there was no statistically significant difference in IRS-1 associated PI3K activity in M-p110 β KO mice, there was a trend towards reduced activity of pTyr-associated PI3K activity in these mice ($p = 0.12$). We have now updated our results with these findings (Results, page 4, line 19,20).

Finally, if the results in Supplemental data 1f and 1g are going to be described as similar to the PI3K activity data, then they should be analyzed, ie mean \pm sem and a statistical comparison, in the same fashion. Otherwise it should be noted that this is a qualitative comparison

We find the PIP3 immunofluorescence is difficult to quantitate, and thus we have noted that PIP3 immunofluorescence images represent a qualitative conformation of PI3K activity data (Results, page 4, lines 20-23).

The statement that increased proteasomal and autophagosomal pathway activities are due to decreased FoxO1 phosphorylation and activity in the results section is an overstatement. This should be described as an association and limited only to the phospho FoxO1 data as there is no direct measure of FoxO1 activity presented. Also, the statement is somewhat contradictory to the observation that there were no differences in proteasome-related gene expression and the authors' own statement that "increased proteasomal activity may be induced by a post-transcriptional or conformational change in the protein." Alternatively, measuring transcriptional readouts of FoxO1 activity that mediate muscle atrophy via the ubiquitin-proteasomal pathway – MuRF1 and atrogin-1 – would strengthen conclusions related to FoxO1.

The reviewer is correct to note that we did not have data on FoxO activity. Since this is an important point, we have further investigated FoxO activity by quantification of nuclear versus cytoplasmic localization of FoxO proteins. In spite of decreased FoxO1 phosphorylation, we were not able to demonstrate increased nuclear localization of FoxO1, 3 or 4. Thus, we have removed our speculation about FoxO1 activity, as suggested by the reviewer (Results, page 7, last paragraph). We replaced phospho-FoxO1 data with new data on FoxO nuclear translocation and have included it as Supplemental Figure S3c. Since we could not demonstrate increased FoxO1 nuclear translocation and found no difference in expression of proteasome related genes, it is unlikely that other targets of FoxO nuclear activity would be altered in M-p110 α KO mice.

There are a number of references that describe how loss in insulin receptor signaling results in FoxO1 mediated skeletal muscle atrophy via the ubiquitin-proteasomal pathway and they should be discussed.

We have now included references to papers on FoxO mediated regulation of ubiquitin-proteasome and autophagosome pathways (O'Neill, JCI. 2016 and Sandri, Cell. 2004; Mamucari, Cell Metab. 2007) and increased discussion of this point as suggested. These studies show that activity of ubiquitin proteasome pathway is markedly increased in atrophying muscle due to transcriptional activation of ubiquitin; several proteasomal subunit genes; and two muscle-specific ubiquitin ligases, atrogin-1/MAFbx and MuRF1, which are induced several-fold during early stages of muscle atrophy. However, since space was limiting, we were not able to discuss these points in detail.

Finally, phospho FoxO1 data is shown in the fasted state and may be more meaningful in fed-state. Nuclear localization by microscopy or cell fractionation would be more convincing than phospho levels alone.

As mentioned above, we have performed cell fractionation experiments in both fed and fasted condition and assessed FoxO1, FoxO3, and FoxO4 protein levels in Lox and M-p110αKO mice. Due to sample to sample variability, there was no statistically significant difference between lox and KO mice, but nuclear FoxO1 localization was increased in both groups with fasting, as expected.

Given what appear to be fiber type specific effects of p110a deletion, particularly the mitochondrial morphology data, it is critical to include a more thorough characterization of the p110a deletion in different muscle depots/fiber types (EDL, soleus, TA, gastrocnemius).

As requested by the reviewer, we have now assessed p110α deletion in different muscle groups. As shown below, we find that p110α KO in TA, pooled hind limb, soleus, and quadriceps have similar level of knockdown. What is interesting is that there is marked upregulation of p110α in quadriceps muscle upon fasting as compared to fed conditions in both genotypes. However, this upregulation with fasting is not observed in mainly oxidative soleus muscle. Thus, it appears that the level of KO is similar in different muscle depots, but p110a protein levels are expressed differently in different muscle types upon feeding and fasting.

Throughout the manuscript different muscle depots are used without rationale for why this was done or why it may not matter and this should be discussed. For example, KO and signaling data in Fig. 1 are from mixed fiber type sources (gastroc and quad), while RNA-seq data in this figure are from a primarily fast/glycolytic depot (tibialis ant). Follow up studies in Fig. 2 exploring the differences in autophagy gene expression from RNA-seq in figure 1 use mixed fiber quadriceps and gastroc for protein measures and proteasome activity, while gene expression is measured from fast/glycolytic tibialis anterior.

The reviewer brings up an excellent point. It appears that the knockdown of p110α has more effect in glycolytic than oxidative muscles. Thus, muscle weight was decreased (Fig S6b), whereas proteasome degradation (Fig 2f) and average mitochondrial area (Fig 3a and b) were increased in glycolytic muscles, but these parameters were not altered in oxidative, soleus muscle. These results are in agreement with our previous publication, showing that muscle-specific loss of insulin signaling created by knockout of IR and IGF1R results in increased numbers of oxidative muscle fibers (O'Neill et al, JCI. 2016). Thus, a loss of insulin action induces fasting-like phenotype and shifts metabolism from glycolysis to fatty acid oxidation, a change which is more noticeable in glycolytic muscle rather than in oxidative muscle, which already has robust fatty acid oxidation. We have now explained this important observation in the revised Results and Discussion sections.

The observation that age-dependent muscle proteolysis (2f), autophagy gene expression (2b) and mitochondrial area (3a) are only different in fast/glycolytic depots (TA and EDL) would argue for focusing here, and may explain some of the inconsistencies in other figures, such as the lack of effect of fasting on ULK1 phosphorylation and the LC3II/I ratio in mixed fiber tissues in 2a and 2c.

The reviewer is correct again. As shown above, LC3 protein levels are differentially modulated in quadriceps muscle, which is composed of mixed muscle fibers, as compared to soleus muscle, which is composed primarily of oxidative fibers. As an example, LC3 levels were increased in quadriceps muscle of M-p110αKO mice as compared to controls, in both fed and fasted conditions. On the other hand, in soleus muscle, there was much less LC3 upregulation in M-p110αKO mice as compared to controls under fed conditions, and no increase what so ever after fasting. These data have been added to supplemental figures (Fig S2f) and described in the Results section.

The fact that fasting does not appear to increase pULK1 or LC3II/I ratio compared with fed samples (control or KO) should be discussed.

As pointed out by the reviewer, fasting is expected to increase flux through autophagy pathway. However, LC3 and ULK1 levels are static representation of this dynamic pathway and do not tell us much about the flux through the pathway. In order to assess flux through the pathway we have now included experiments in which we treated mice with colchicine to inhibit protein degradation. Following this treatment, ULK1 and LC3 protein levels were indeed increased in control and M-p110αKO mice in the fasted state. This data are now included in a new Fig 2a and described in the Results section.

GTTs were performed after an overnight fast and this may account for the lack of difference as fasting of this duration is insulin sensitizing in the mouse. Relative hyperinsulinemia in the fed-state suggests some degree of insulin resistance. The conclusion that there is no difference in glucose homeostasis would be more convincing if additional studies were performed. A short term fast prior to GTT is suggested and may reveal impaired glucose tolerance, as suggested by the fed plasma insulin levels and would be more physiologically relevant. Measuring plasma insulin during the GTT minimally at the 0 and 30 min time points may reveal insulin resistance/compensatory insulin secretion.

We have now performed GTT after a short 4h fast (10 am to 2pm) in 6-month old mice. Even under these conditions there was no difference in glucose homeostasis between M-p110αKO and control mice. We have also measured insulin levels during GTT at 0 and 30 min time points as suggested by the reviewer. M-p110αKO mice tended to have increased insulin levels as compared to the controls at 0 min, but not at 30 min time point, but these differences between control and KO mice did not reach statistical significance. These data are similar to the data already shown in Fig 6a,b,c and thus are provided here to the reviewer, but have not been included in the manuscript.

Ideally a hyperinsulinemic euglycemic clamp could be performed to resolve whether p110a deletion in muscle impairs insulin-stimulated glucose disposal.

It was not feasible to perform hyperinsulinemic euglycemic clamp study, but we have performed *in vivo* glucose uptake utilizing radioactively labeled glucose in 6 month old mice. Interestingly, there was no difference in insulin stimulated glucose uptake in tibialis anterior and soleus muscle in spite of decreased insulin signaling upon M-p110αKO deletion. This is in line with our data showing that there is no difference in glucose tolerance and insulin sensitivity in these mice. These data are now included in Fig S5f and described in the Results section.

While initial characterization of the muscle specific insulin receptor KO (MIRKO) mice demonstrated no change in glucose tolerance, subsequent clamp studies in these mice demonstrated impaired whole-body insulin sensitivity due to impaired skeletal muscle glucose uptake (Kim JK, JCI, 2000. PMID 10862794). These data should be discussed in the context of this model, as well.

We have now discussed decreased glucose uptake in MIRKO mice, as detected by hyperglycemic euglycemic clamp test.

Another modest oversight in the manuscript is the lack of discussion of work from Lewis Cantley's group looking at the effects of p85 subunit deletion and subsequent effects on glucose tolerance and muscle mass. Deletion of p85 isoforms is reported to reduce p110 protein levels as well, resulting in similar reductions in muscle mass, but contradictory impaired glucose tolerance compared to data reported here (Luo J, Cell Metab, 2006. PMID 16679293). A more complete comparison of the p110a MKO mice to MIRKO and other skeletal muscle Akt or p85 null models is warranted in the discussion, specifically with regards to changes in muscle mass, mitochondrial mass and glucose intolerance/insulin resistance.

As reported by Cantley's group, muscle specific p85a deletion resulted in decreased p110a levels but these mice also did not develop impaired whole body glucose tolerance. It was only when muscle p85a KO mice were crossed with whole body p85b KO mice that glucose intolerance could be detected. We have now included a discussion of these findings in our revised Discussion section as suggested by the reviewer.

Muscle weight should be expressed as a percentage of body weight, as is done for fat mass, or per tibia length. This will distinguish whether muscle mass is specifically reduced or if these are smaller mice in general, which the reduced body weight suggests.

Once adjusted for body weight, there was no difference in muscle weights between M-p110 α KO and control mice. Since body weight is reduced, as muscle weight decreases, adjusting one variable with another dependent variable abrogates this difference. However, M-p110 α KO mice are not smaller than control mice, as there is no difference in femur length between genotypes. Thus, we believe that the decreased body weight in M-p110 α KO mice is primarily driven by decreased muscle mass and, to some extent, decreased fat mass. We believe that our original representation of the data as absolute muscle weight more accurately describes the physiology than muscle weight as a percentage of body weight.

The statement in the results that increased skeletal muscle triglyceride is due to increased FATP1 expression is an overstatement. A more balanced interpretation is that increased triglyceride was associated with increased FATP1. An alternative interpretation is that reduced insulin stimulated

glucose transport due to p110a deletion produced a starvation like effect in muscle where increased reliance on fatty acids was a feature, such that FATP1 expression was induced.

The preponderance of data in our manuscript suggests that a loss of p110a in muscle results in a metabolic switch from glucose utilization to fat metabolism. Increased muscle triglyceride and FATP1 expression is thus a reflection of this metabolic switch to fasting-like phenotype as the reviewer suggested. We have now updated our discussion section to include that a loss of p110a in muscle leads to metabolic switch from glucose to fat metabolism so that primarily glycolytic muscle fibers are more severely affected.

Food intake should be normalized to lean mass for consistency with all other metabolic cage data reported instead of per mouse.

We have now normalized food intake per body weight, and we do not see differences between the genotypes. These data are included in our Results section Fig S7j.

Minor:

Typo on page 4, third line from the bottom where “complete loss of PI3K staining” should read “complete loss of PIP3 staining.”

We thank the reviewer for noticing this misspelling; this has been corrected.

Data in S2a is described in the results as quantification of pAkt, but shows p4EBP1 data.

We regret this mistake, which has now been corrected. pAKT quantification is shown in Fig 1c and p4EBP1 is shown in Fig S2b.

The units in Fig. 6h seem incorrect – mg triglyceride per gram tissue or protein are typical, not a concentration of triglyceride (mg/dl) as is used here.

The units in Fig 6h have now been corrected.

Reviewer #2 (Remarks to the Author):

In this paper the authors investigate the role of the catalytic subunit of PI3K, p110a in skeletal muscle. In line with previously reported observations in liver and adipose tissue, the authors show that in muscle only the loss of p110a, and not p110b, perturbs the PI3K-AKT cascade and therefore results in impaired insulin signaling. The phenotypes have been analyzed after different nutritional challenges (fasting or HFD) and comparing effects observed in young and old mice. Although obesity and diabetes are usually associated with mitochondrial dysfunction, the authors show in this work that increased mitochondrial respiratory capacity, associated to an induction of PGC1a expression in mice lacking p110a in skeletal muscle. Phenotypically, reduced AKT pathway activation is associated with decreased muscle mass and function.

The phenotyping of these mice at the organismal level is complete and thorough and allows the authors to clearly demonstrate the mild nature of muscle dysfunction. Unfortunately, the authors have a tendency to oversell the phenotypic relevance of this mouse model as it clearly does not reproduce human diabetes: rather it is a mild model of starvation-like phenotype. It should be made clearer in the discussion that the increase in mitochondrial respiration might be a putative compensatory mechanism that, together with increased activity of other organs, prevents the development of the disease. Neither of these hypotheses have been experimentally tested and thus the study, while interesting, remains largely descriptive. I feel that the manuscript warrants further revision prior to publication.

We thank the reviewer for her/his thoughtful comments and helpful suggestions. We are greatly thankful for suggesting that a loss of p110a in the muscle leads to mild fasting-like phenotype. We also appreciate his/hers detailed comments and for identifying even slight spelling mistakes.

The phenotyping of these mice at the organismal level is complete and thorough and allows the authors to clearly demonstrate the mild nature of muscle dysfunction. Unfortunately, the authors have a tendency to oversell the phenotypic relevance of this mouse model as it clearly does not reproduce human diabetes: rather it is a mild model of starvation-like phenotype. It should be made clearer in the discussion that the increase in mitochondrial respiration might be a putative compensatory mechanism that, together with increased activity of other organs, prevents the development of the disease. Neither of these hypotheses have been experimentally tested and thus the study, while interesting, remains largely descriptive. I feel that the manuscript warrants further revision prior to publication.

We thank the reviewer for making this point. After generating additional data described above and below, we agree with the reviewer that mitochondrial phenotype and increased fatty acid oxidation is likely a compensatory mechanism secondary to a loss of insulin signaling mainly in glycolytic muscle fibers. We have made this clear in our revised Discussion section.

Major comments:

The quality of the immunoblots in Fig. S2 makes it difficult to draw clear conclusions and is somewhat problematic with respect to P-GSK and P-4EBP where some bands are rounded, and P-FoxO that seems overexposed.

We have repeated western blotting (WB) for p4EBP1 and total 4EBP1 shown in Fig S2d, which now does not show any rounded bands. The new WB was quantified, and the data are consistent with previously shown results. The new WB and quantification are included in Fig S2d and shown below.

We have also repeated pAkt, Akt, pGSK and GSK western blots previously shown in Figure S2e and replaced it with new WB and protein quantification data, which is also shown below. As expected, there was a decrease in Akt phosphorylation between chow and HFD fed control mice, but there was no statistically significant difference in Akt phosphorylation between Lox and M-p110βKO mice on HFD. Performing additional western blots otherwise did not change our original results.

Fig. 1b and S2c should be shown in one panel in order to conclude about the effect of HFD.

For technical reasons we are not able to perform pAKT and pGSK quantification in chow and HFD fed control and M-p110αKO mice on a single western blot, and combining Fig 1b and S2c could be misleading due to different exposures of WBs. However, we have now run pAkt and pGSK blots in chow and HFD fed control and M-p110βKO mice and show decreased Akt phosphorylation in control mice on HFD, as expected.

Fig. S2c, the text refers to ERK but the figure show GSK, which one is correct?

We thank the reviewer for pointing this out. Figure S2c is now correctly labeled to show pERK and tot ERK blots.

Fig. 2a, no effects of fasting in control mice were observed, which is surprising? What is the explanation? It would be helpful for the authors should provide the autophagy markers and the phosphorylation status of AKT and its targets.

Protein turnover increases with fasting and likely confounds our data, so that no observable increase in ULK and LC3 levels was observed with fasting. We have treated control and knockout mice with colchicine to inhibit protein turnover and indeed we see an increase in ULK and LC3 levels upon fasting. Figure 2a has been updated to show colchicine treated mice.

Fig. 2c, the image does not appear to be representative of the differences that are reported by quantification. Fibers are clearly different and it is difficult to understand the scoring of LC3 positivity. Moreover, these data appear incongruent with the protein levels illustrated by immunoblot (Fig. 2a) and the transcript levels (Fig. 2b) where the differences between KO and control mice are mainly in

fed condition

The reviewer is correct to notice that the images in Figure 2c were not the best representatives of the data. We have now included representative images at a lower magnification where the difference in LC3 staining is much more apparent. The previous high magnification images only showed several muscle fibers, which can easily lead to selection bias. Low magnification images clearly show that LC3 staining was increased in M-p110 α KO mice as compared to the controls, which is in agreement with Figures 2a and 2b. This is also shown in images below.

The mitochondrial morphology and structure revealed by electron microscopy show clear differences Fig. 3c however it is unclear whether these are cause or consequence. Could it be that fibre disruption (see Figure 2c) leads to downstream mitochondrial perturbations? Differential transfection capacity of mutant and wild type muscle fibers could explain the observed differences. The mitochondrial morphology images provided in Figure 3c are unconvincing and it is unclear how one can define mitochondrial area using these data.

As the review pointed out in the opening paragraph, mitochondrial changes observed in M-p110 α KO mice are likely secondary to a decrease in insulin signaling resulting in a starvation-like phenotype that relies on increased mitochondrial fatty acid oxidation. Muscle fiber size does decrease in M-p110 α KO mice (Fig 6d), so the profound difference in mitochondrial morphology reported in Figure 3c is, in part, confounded by a decrease in muscle fiber size. Total muscle area in Lox mice is much larger than in KO mice (Fig 3c), thus increased mitochondrial density in M-p110 α KO mice is a product of increased mitochondrial mass per muscle fiber (Fig 3a), but also of increased muscle fiber density due to smaller fibers. We have now made this point clear in the Results section that the increase in total mitochondrial area reported in Figure 3c is amplified by an increase in muscle fiber density in M-p110 α KO mice.

Moreover, I call into question the relevance of mitochondrial dynamics in this model and doubt that the mild alterations in steady state levels of the fusion and fission proteins reported in Fig. 3f are sufficient to alter mitochondrial morphology directly.

As previously reported, mitochondrial fission is sufficient to cause muscle wasting (Romanello V et al., The EMBO Journal 2010). While mitochondrial dynamics clearly have an effect on muscle remodeling, other pathways such as autophagy/mitophagy and substrate availability also contribute to the observed phenotype.

Moreover, mitochondrial morphology in muscle fibers appear similar between mutant and wild type in Fig 5f. How can the data in this figure and Fig 3f be reconciled?

Figure 5f shows increased pMitoTimer quantification of ROS in M-p110 α KO mice, but it also shows more elongated mitochondria in these mice. This is in line with the data reported in Figure 3f, showing increased levels of proteins involved in mitochondrial fusion in M-p110 α KO as compared to control mice.

DRP1 activity is depending on the balance in the phosphorylation of two different serine residues, authors should provide the status of P-DRP1 (S616). If increased fusion is at play, attempts to reduce fusion capacity either by downregulating OPA1 or both Mitofusins (both of which are reportedly upregulated) would be predicted to rescue the phenotypes. Indeed, muscle-specific deletion of OPA1 has been demonstrated to affect insulin signaling and metabolism and thus it would be interesting to know how the authors data fit into this paradigm.

We have made several attempts to quantify DRP (S616) phosphorylation. Unfortunately due to poor antibody quality, we were not able to generate these data.

Reduced OPA1 levels have been observed in humans with insulin resistance. Increased OPA1 levels in M-p110 α KO mice likely represents a compensatory mechanism by which muscle cell undergoes a shift from glucose to fat oxidation. This paradigm has now been stated in several segments of our manuscript, as suggested by the reviewer.

In Fig. 6e and S6g, the data do not support the conclusion that after fasting the increase in myogenetic markers are further exaggerated.

The reviewer is correct. We have now modified our statement to read: "The decrease in myostatin and increase in MyoD1 were also observed even after 24h fasting (Fig. S5h), when protein catabolism of muscle is increased."

In Fig. 6f, the authors discover reduced circulating myostatin levels and argue that it is associated with increased muscle mass. However, this mouse model exhibits reduced muscle mass, accompanied by increased % of central nucleated fibers and increased expression of myogenetic markers and fibers number, but also increased TG accumulation and decreased fiber area. A proliferation assay would help to better clarify this phenotype

We thank the reviewer for noticing this important distinction. While circulating myostatin levels were decreased and expression of genes regulating muscle growth were increased, this is likely a compensatory mechanism in response to decreased muscle mass in M-p110 α KO mice. This point has now been clarified in our Discussion section.

Minor comments:

_ 3rd paragraph of the introduction: de ceased instead of decreased

We have now corrected this spelling mistake

_ Reference to Fig. S1c is missing

We have now described Figures S1b and S1c, which report p110a mRNA and protein levels in control and M-p110 α KO mice

_ to be able to compare the effect obtained in the p110a KO compare to the lack of effects in the p110b KO, Fig. S1f and S1g should be set to the same level of signal intensity

This correction has been made as instructed.

_ Fig. S2e, authors should confirm that the increase P-AKT in p110b KO mice is not statistically significant under CD

This figure has been replaced by a new Fig S2e, in order to improve western blot quality as requested by this reviewer. There was no difference in AKT phosphorylation between M-p110bKO and control mice in either old or new version of the figure.

_ Fig. S3b, the age of mice in figure legend does not correspond to the one specified in the text

The experiment was done at 5 month of age as stated in the figure legend. The text has been corrected accordingly.

_ The reference to Fig. 3a and 3b should go after next sentence

We have moved the order of sentences in this paragraph to describe the changes in M-p110aKO mice first, before describing the changes in control mice.

_ Fig. 3f, authors should confirm that the increase P-DRP1 (S637) in Lox mice is not statistically significant upon fasting

We have re-quantified the WB for p-DRP1, and there was no statistically significant increase in Lox mice upon fasting. New figure is shown below and included in the Results section.

_ Fig. S4e (or Fig. S7a), given the results obtained analyzing the expression of PGC1 α , it would be interesting in these screening to include PPAR α .

Unfortunately we have not quantified PPAR α protein levels in these mice. However, in Figure S6k we show that PPAR α expression was indeed increased in adipose tissue of M-P110aKO mice in line with increased PGC1 α protein levels.

_ Fig. 5d, the levels in fed mice should be shown as reference

Unfortunately we have not quantified NADH autofluorescence in fed mice.

_ Fig. 5e, no correspondence between text, graph and figure legend, is it ratio ROS/RNS?

The assay that was used measures total free radical activity, which is composed of both ROS and RNS. The Figure legend has been updated with this additional information. More information about the assay can be found here:

<https://www.cellbiolabs.com/sites/default/files/STA-347-in-vitro-ros-rns-assay-kit.pdf>

_ Fig. 5f and S4f, figure legend should be corrected with the proper muscle type

As described in both figure legends pMitoTimer plasmid was transfected in situ into quadriceps muscle fibers of living mice.

_ Fig. S5a, the HFD treatment duration indicated in figure legend does not correspond to the one specified in the text

The mice were on HFD for 13 weeks. The figure legend has been corrected accordingly.

_ Fig. S6g, correct myogenetic in the text

This has been corrected.

_ Fig. 6h, the age of the mice used for the TG measurement should be specified

TG was measured at 3 months of age. This information is now added to the figure legend.

_ Fig. S6, it would be interesting to include in the analysis TG levels and lipogenesis status in liver.

There was no difference in liver TG levels in these mice as shown below.

Reviewer #3 (Remarks to the Author):

The study from the Kahn lab relates to the role of the catalytic PI3K isoforms in skeletal muscle. By various studies in animals they decipher a positive role of the p110alpha (rather than beta) isoform in metabolic action and muscle mass (incl. inhibition of protein degradation). Yet, the muscle of the transgenic p110 alpha KO animals displays enhanced mitochondria capacity/ function. A phenotype proposed to be mediated by elevated PGC1a expression and hypothesized to result in enhanced NADH content/production and free radical release. The author, conclude that the p110 alpha isoform is a key controller in the insulin induced regulation of metabolism, muscle mass and mitochondria homeostasis.

The amount of data provided in this investigation is overwhelming and in full support the ideas put forward by the authors. Also, the study / analyses and methodology applied are of high quality. So in many ways this is a trustful study providing new interesting insights.

Having this huge capacity and skills (Kahn lab), I would have liked to see the authors pushing the ideas to the limits at which the model proposed do not hold as tight or at all. It would be beneficial for the whole society if these limitations become to display – also in journals like Nat Comm. So let us hear the critical voice in this study – telling the reader where – in the view of the authors – the limitations are.

Below I have put forward some thoughts that could be considered:

We thank the reviewer for providing valuable comments and greatly appreciate his or her remarks regarding the role of PI3K in myostatin expression.

Below I have put forward some thoughts that could be considered:

If myostatin is key to the phenotype (as proposed in the discussion) – it might be of interest to explore whether PI3K is a key player in myostatin expression/release or receptor expression?

Previously, our lab has generated heterozygous knockin mutation in *Pik3r1* gene, which results in Arg 649 to Trp substitution. This dominant negative mutation impairs insulin signaling when expressed in fibroblasts. Similar to the effect in M-p110aKO mice, this mutation was also associated with decrease in myostatin expression and a tendency for increased MyoD1 levels. These data further support our evidence that impairment of insulin signaling through manipulation of PI3K subunits leads to decrease in myostatin levels. These data are now included in our manuscript (Fig S6f) and described in the Results and Discussion sections.

The idea that PGC1a-> increase mito-function -> increase ROS production -> increase autophagy,- is interesting and well supported by the data. I recall that the antioxidant capacity has also been positively associated to enhance PGC1a expression. Was this evident here as well?

We certainly see that some genes that regulate ROS levels such as glutathione S-transferase (GST) were increased in M-P110aKO mice, but others such as catalase and NAD(P)H Quinone Dehydrogenase 1 were not altered. We provide these data for the reviewer but did not include them in the manuscript due to length considerations and the abundance of data already present.

How well is the decreased muscle function related to the loss of muscle mass. Clearly the endurance is unrelated to the mito-function/capacity.

As shown below, in addition to exercise capacity, grip strength was not reduced in M-P110aKO mice at 3 months of age. Thus decreased muscle function cannot be solely explained by the loss of muscle mass.

Somewhat ignored in the paper is the interesting observation that the phenotype observed in white muscle is not copied in the red- type II rich – muscle. Why is this? Here is an interesting observation that might point to some interesting mechanisms. Also, 50% of human muscle mass is composed of type I fibers – so the observation in soleus may also relate to human-translational aspects.

The reviewer brings up a valid point which was also raised by reviewer #1. M-p110 α KO mice have a decrease in insulin signaling, which results in starvation-like phenotype and a shift from glucose utilization to fatty acid oxidation. Since glycolytic muscles naturally rely more on glucose utilization, the phenotype is stronger in white muscle rather than in red muscle, which already has robust fatty acid oxidation. As suggested by the reviewer, this interesting point is discussed in detail in a new paragraph of the discussion section.

The authors relate to possible association between insulin action and mitochondria function. Although congenital insulin receptor signaling defect has been shown to lead (associate) to mitochondria dysfunction in man (e.g. Kristensen et al: PMID: 24510228 DOI: 10.1007/s00125-014-3187-y), the causality of these parameters is highly controversial in man mainly because of poor matching between cohorts (e.g. T2DM and controls) – in particular- to the main determinant of mitochondria biogenesis (physical activity). The mouse model presented here may provide some further insight to this association (although opposite to the above human findings) if used in a more detailed temporal manner. Alternatively, an inducible KO model may be of value also. Thus, - for me -deciphering the temporal aspect of development for the phenotype could add huge to understanding of any causality. I do not ask the authors to add in a new model – but some thoughts about this would be of value for the paper.

We agree with the reviewer that temporal relationship between insulin resistance and mitochondrial dysfunction is important, but it is not practical to repeat all experiments in the inducible IR knockout model. While mitochondria in M-p110 α KO mice are certainly larger and have increased mitochondrial complex activity, they also produce more ROS and thus are contributing to increased muscle turnover and reduced muscle mass. This interesting point is now emphasized in our discussion.

It is interesting that the signaling defect only lead to moderate insulin resistance. The Kahn Lab has previously shown that physical activity is a major confounder for the phenotype of the MIRKO mouse model (PMID: 10545524 PMID: PMC409827 DOI: 10.1172/JC17961). I wonder if the p100 α KO phenotype is also affected by the lesser level of physical activity.

We have measured physical activity in these mice at 3 months of age and did not see decreased physical activity in these mice. We provide these data for the reviewer but ask not to include it in the manuscript due to length considerations.

I believe that several papers have shown that PI3K inhibition by wortmannin and the LY compound affects both the Akt- and the Erk signaling partway in muscle – and thus I do not understand the authors are surprised by this page 5 (top)

We thank the reviewer for pointing this out. We have now corrected our miswording in the result section.

Reviewers' Comments:

Reviewer #1:

Remarks to the Author:

The authors have addressed my concerns and now provide a more balanced interpretation of their results, particularly the relationship between insulin signaling, glucose homeostasis, autophagosomal/proteasomal activity and mitochondrial homeostasis. The additional experiments performed to satisfy questions related to fiber type-specific effects, autophagic flux and glucose transport are commendable. Although the phenotype of the p110a MKO mice is largely predictable based on this groups' previous work and the work of others, the breadth and number of experiments performed and the novel observation that p110a is the dominant p110 subunit in skeletal muscle increase the impact of the manuscript and it should be of great interest to the field.

Reviewer #2:

Remarks to the Author:

The authors have made several important revisions to the manuscript in order to address concerns raised by the reviewers. Importantly, they have also toned down correlative overstatements. Additional elements in the discussion were also added that provide context for what was already known regarding PI3K, muscle metabolism and function in the mouse. The authors have also made significant efforts to increase the quality of their data by repeating western blots and providing more representative micrographs, which help strengthen their claims and the manuscript in general.

I still remain unconvinced by Figure 3c and the determination of mitochondrial area in transfected fibres. These claims of increased mitochondrial area based on these data are consistent with Figure 3a/b, (TEM) which is of much higher quality and would recommend showing only Figure 3a/b in the main figure. The general notion that mitochondrial biogenesis is augmented in the KO muscle due to PGC1 induction is supported by Figure 3a/b, f and Figure 4. The question now is how alteration in PI3K leads to the transactivation of PGC1 and whether abolishing PGC1 (and the putative secondary mitochondrial compensation) would reveal a more profound metabolic and functional phenotype in these mice. This however, is clearly beyond the scope of the current study.

Nevertheless, the manuscript is now, in my option, acceptable for publication.

Reviewer #3:

Remarks to the Author:

Dear authors

I think you have responded in a fair and competent manner to my suggestions and ideas. I have no further comments.

* Your paper will be accompanied by a two-sentence Editor's summary, of between 250-300 characters including spaces, when it is published on our homepage. Could you please approve the draft summary below or provide us with a suitably edited version.

Diabetes is associated with decreased PI3K activation in skeletal muscle. Here, the authors show that p110alpha is the predominant PI3K subunit in muscle and show that ablation of p110alpha in muscle, but not ablation of p110beta, leads to insulin resistance, increased proteosomal and autophagic activity, and altered mitochondria homeostasis in mice.

- We approve the revised draft summary above.

REVIEWERS' COMMENTS:

Reviewer #1 (Remarks to the Author):

The authors have addressed my concerns and now provide a more balanced interpretation of their results, particularly the relationship between insulin signaling, glucose homeostasis, autophagosomal/proteasomal activity and mitochondrial homeostasis. The additional experiments performed to satisfy questions related to fiber type-specific effects, autophagic flux and glucose transport are commendable. Although the phenotype of the p110a MKO mice is largely predictable based on this groups' previous work and the work of others, the breadth and number of experiments performed and the novel observation that p110a is the dominant p110 subunit in skeletal muscle increase the impact of the manuscript and it should be of great interest to the field.

-Thank you for the positive feedback from reviewer #1.

Reviewer #2 (Remarks to the Author):

The authors have made several important revisions to the manuscript in order to address concerns raised by the reviewers. Importantly, they have also toned down correlative overstatements. Additional elements in the discussion were also added that provide context for what was already known regarding PI3K, muscle metabolism and function in the mouse. The authors have also made significant efforts to increase the quality of their data by repeating western blots and providing more representative micrographs, which help strengthen their claims and the manuscript in general.

I still remain unconvinced by Figure 3c and the determination of mitochondrial area in transfected fibres. These claims of increased mitochondrial area based on these data are consistent with Figure 3a/b, (TEM) which is of much higher quality and would recommend showing only Figure 3a/b in the main figure.

-Thank you for the comments from reviewer 2. We have removed Figure 3c and all the other relevant information accordingly.

The general notion that mitochondrial biogenesis is augmented in the KO muscle due to PGC1 induction is supported by Figure 3a/b, f and Figure 4. The question now is how alteration in PI3K leads to the transactivation of PGC1 and whether abolishing PGC1 (and the putative secondary mitochondrial compensation) would reveal a more profound metabolic and functional phenotype in these mice. This however, is clearly beyond the scope of the current study.

Nevertheless, the manuscript is now, in my option, acceptable for publication.

Reviewer #3 (Remarks to the Author):

Dear authors

I think you have responded in a fair and competent manner to my suggestions and ideas. I have no further comments.

--Thank you for the positive feedback from reviewer #3.